# High-resolution operational soil moisture monitoring for forests in the Middle Germany

Ivan Vorobevskii[1], Thi Thanh Luong[1], Rico Kronenberg[1], Rainer Petzold[2]

[1]Faculty of Environmental Sciences, Chair of Meteorology, TUD Dresden University of Technology, Tharandt, 01737, Germany
[2]Competence Centre for Forest and Forestry, Saxony Forest State Enterprise, Pirna, 01796, Germany

*Correspondence to*: Ivan Vorobevskii (ivan.vorobevskii@tu-dresden.de)

**Abstract.** The forests of Central Germany (Saxony, Saxony-Anhalt, and Thuringia) are vital components of the local ecosystems, economy, and recreation. However, in recent years, these forests have faced significant challenges due to prolonged climate-change-induced droughts, causing water shortages, tree stress, and pest outbreaks. One of the key components of the forests' vitality and productivity is the availability of soil moisture. Given the anticipated increase of frequency and severity of droughts events, there is a growing demand for accurate and real-time soil moisture information. This underscores the need for development of an appropriate monitoring tool to make forest management strategies more effective.

The article introduces an operational high-resolution soil moisture monitoring framework for the forests in Middle Germany. The key components of this system include advanced LWF-BROOK90 1D water balance model, large database of National Federal Forest Inventory, high-resolution forest soil maps, real-time climate data from German Meteorological Service, and web information platform for the presentation of daily updated results. This system informs the public and empowers forest managers and other decision-makers to take targeted, local measures for sustainable forest management, aiding in both drought mitigation and long-term forest health in the face of climate change. The validation of the system using soil moisture measurements from 51 stations with various sensor depths (up to 100 cm) showed an overall good agreement (0.76 median Pearson correlation), which was found higher for deciduous rather than for coniferous forests. Finally, the framework is discussed against the background of main limitations of existing monitoring systems and how operational soil moisture measurements contribute to the better interpretation of simulations.

## 1 Introduction

Forests of Central Germany (Saxony, Saxony-Anhalt and Thuringia) are of ecological, economical and recreational importance in the region. In recent years, climate changes, particularly prolonged droughts in 2018, 2019, 2020 and 2022 (Meusburger et al., 2022; Obladen et al., 2021; Patacca et al., 2023; Spiecker and Kahle, 2023), and have significantly affected the forest ecosystem. These droughts not only directly led to water shortages and associated stress for trees and thus the whole forest ecosystem (Buras et al., 2020), but have caused indirect damage through pest outbreaks, such as the bark beetle (Hlásny et al.,

2021). Moreover, it was found that especially under drought conditions, the weakened trees are less resistant to such pests (Vindstad et al., 2019). From 2018 until 2022, pest calamity and droughts have caused a loss of 500 000 hectares of forests in Germany, demanding for 900 million euros for climate-adapted forest management (Bundesministerium für Ernährung und Landwirtschaft, 2023). The quick spread of the insect infestation and other associated (and non-associated) forest damage (i.e.

windfall) combined with the potential increase of the drought stress due to climate change highlight the urgency of the situation and the need to develop effective countermeasures for both short- and long-term management strategies (Albrich et al., 2020; Schuldt et al., 2020).

The vitality and productivity of forests are highly dependent on the amount of water available in the soil (Spiecker and Kahle, 2023; Zang et al., 2014). Under changing climate conditions, drought events and extreme weather conditions are likely to

become more frequent and severe (Hanel et al., 2018; Orth et al., 2016; Zhou et al., 2019), it is essential to have reliable estimations of soil water availability in forests (Meusburger et al., 2022). Up-to-day soil moisture information influences a wide range of forestry decisions from routine forest management and risk assessment to specific mitigation measures on sensitive sites (Sharma et al., 2022; Zweifel et al., 2023). For example, it plays a major role in planning of the planting and trimming actions based on current soil moisture conditions (Scholz et al., 2023). In addition, this real-time data helps with risk

management, such as assessing the risks of wildfires during dry periods, the growth of associated pathogens, treefall of stressed and vulnerable trees in wet periods. In ecologically sensitive forest areas, operational data helps to make an appropriate choice of specialised technologies to maintain forest health and resilience. Thus, a precise real-time monitoring system of soil moisture can offer not only effective measures to mitigate drought vulnerability of forests, but also a crucial tool for sustainable forest management in the future.

Currently, few initiatives exist, such as for example North-western Switzerland (https://www.bodenmessnetz.ch) and Cosmic-ray soil moisture network in UK (https://cosmos.ceh.ac.uk) on a local national scale and the International Soil Moisture Network (https://ismn.earth/en/dataviewer, Dorigo et al., 2021) on a global international scale, which provide data on in-situ point-based soil moisture measurements. However, all these networks are still limited by their spatial and temporal coverage due to the very high operational costs. In this context, site-specific operational water balance modelling, especially in

combination with climate change scenarios, is gaining attention over the measurement-based information platforms. At present, two nationwide platforms exist for this purpose in Germany. The German Drought Monitor (https://www.ufz.de/index.php?en=37937, Zink et al., 2016) has a resolution of 4 km and demonstrates simulation results from Mesoscale Hydrological Model, statistically converted to a specific soil moisture index representing moisture anomalies for topsoil and dull soil column, which however are not accounting for different forest types. The German Meteorological Service

Soil Moisture Viewer (https://www.dwd.de/DE/fachnutzer/landwirtschaft/appl/bf_view/_node.html) utilises AMBAV agricultural and LWF-BROOK90 hydrological models parameterized for 1 km grid size and shows the plant-available water for three types of short vegetation and four tree species. However, despite the big potential of these systems for national-scale soil moisture assessment, they possess certain limitations in terms of their precision and interpretation scope. Despite advances in the development of process-oriented hydrological models for the forests and the availability of the high-resolution datasets

for their parameterisation in recent years (Hoermann and Meesenburg, 2000), these systems do not fully utilise their potential. Furthermore, they lack the ability to account properly for the small-scale variability of soil, land use and weather factors.

This motivates us to come up with a high-resolution point soil moisture monitoring system that is updated daily and takes into account important local site information such as for example aspect, slope, soil type and its profile. Thus, this site-specific data allows forest managers and decision-makers to take targeted local-scaled measures for sustainable forest management. In

addition, a daily update enables timely detection of anomalies or emerging droughts, allowing for quick adaptation measures. The decisive value of our approach stems from the integration of existing inventory systems, in particular the Federal Forest Inventory, which provides detailed, site-specific and integrated information on the forests of Central Germany. Combining these datasets with an operational meteorological data, the effects of different tree species on the changes in soil moisture for a local scale can be better understood.

In this paper, we present an operational soil moisture monitoring system for Middle Germany that addresses the shortcomings of existing monitoring products and uses the current state-of-the-art forest hydrological modelling techniques, providing results in real-time for forest practitioners or interested users using a web-platform. We focused on a detailed and transparent technical description of the system architecture and the data representation on the website. Special attention is given to the qualitative and quantitative analysis of the operational meteorological data used for the model forcing.

**2 Methods and data**

**2.1 Middle Germany region**

Historically, the Middle Germany region (Fig. 1) covers three states (approximately 55000 km$^2$) – Saxony (SN), Saxony-Anhalt (SA) and Thuringia (TN). The topography of the region ranges from the lowlands (0-200 m) in the north of SN and SA to mountainous regions on the south of SN (up to 1200 m) and TN (up to 1000 m). Prominent elevated geographical features

are the Ore Mountains on the border with the Czech Republic, the Harz Mountains in the western part of SA and Thuringian Forest in the south of TN.

The climate conditions of Central Germany are characterised by a moderate continental climate. According to Köppen-Geiger climate classification (Kottek et al., 2006), all three states have predominantly Dfb (hemiboreal) climate type, meaning warm summers and cold winters, with occasional heat spells and typical frost-free periods of 3-5 months. Mean annual air

temperature increases from +3-5$^o$C in southern elevated parts to +10-12$^o$C in the flatlands. Pronounced annual cycles introduce a high variability between summer and winter months as well as in between day-night temperatures. Maximum daily values could reach up to +35$^o$C, while minimum daily temperatures can go far below zero (up to -21$^o$C in Ore Mountains). Due to orographic lifting effects, variations in the annual precipitation amounts are high as well. Mountainous regions of TN and SN receive 1100-1500 mm annually, while lowlands in the north get only 500-700 mm per year. Typically, around 70% of annual

precipitation falls from May to September and the driest month is October.

According to Copernicus Global Land Service: Land Cover 100m (Buchhorn et al., 2019), total forest coverage of the region is about 36.9% (Fig. 1), from which evergreen needleleaf forests prevail and occupy 17.7%, deciduous broad leaf forests cover 7.6% and the rest are mixed forests (11.6%). The coniferous needle leaf forests are dominated by Norway spruce (Picea abies), Scots pine (Pinus sylvestris) and European larch (Larix decidua). The deciduous forests are dominated by European beech (Fagus sylvatica) and pedunculate oak (Quercus robur).

Soil types in the forests of Central Germany are highly variable depending on the topography, underlying geology and forest type. In general, in the mountainous regions of TN and SN podzols and brown soils are common for the forests and in lowlands gleys (or pseudogleys) appear along with abovementioned types as well (Krug, 2000). Typically, forest soils are characterised by rich humus horizons resulting from decades of accumulation of deciduous and coniferous litter. In deciduous forests, especially those dominated by beech and oak, the soils are often loamy with high humus content that promotes fertility. Coniferous forests, especially those dominated by spruce, often have pale podsolic loamy (or sandy loam) soils. These are typically more acidic and much less fertile than in deciduous forests. Mixed forests could combine the soil properties of deciduous and coniferous forests, although the specific properties are highly variable depending on the dominant tree species and site conditions.

## 2.2 Forest monitoring data

### 2.2.3 Data from the German National Federal Forest Inventory

The National Federal Forest Inventory (NFI) is a long-term national German project aiming to collect and store information about forested areas in the country ('Bundeswaldinventur' or 'BWI' in German). It not only provides a comprehensive overview on the condition of Germany's forests, but also integrates important soil information. Every ten years, field observations are conducted to record tree species changes, growth data, update soil profile data and other relevant forest and soil information. To allow intercomparison between inventories, observations are made on the same plots and using standardised procedure. Due to the secrecy of real plot coordinates, NFI rounds coordinates of the plots for the publishing, thus forming square/triangle-formed shapes (tract corners, further referred as NFI sites) with a side of 1 km within which the real inventory plot(s) lie. The third round of the NFI, which data is open sourced and was used in this study, took place in 2012 (https://www.bundeswaldinventur.de, Bundesministerium für Ernährung und Landwirtschaft, 2014). The fourth round started in 2021 and was completed by the end of 2022, however data processing is not finished yet (publication is planned for the end of 2024). The data provides insights into forest management, tree species composition, timber use and soil conditions. In particular, the soil data collection and update are essential for determining nutrient availability and water holding capacity, which directly affect forest health and growth. They also help to assess carbon sequestration and are key indicators of forest health and resilience to environmental stress. By linking forest and soil data with climate and site information, profound analyses of the effects of different environmental factors on forests can be carried out. An overview of the selected NFI sites, as well as summary of the dominant tree species are presented on Figure 1 and in Table 1.

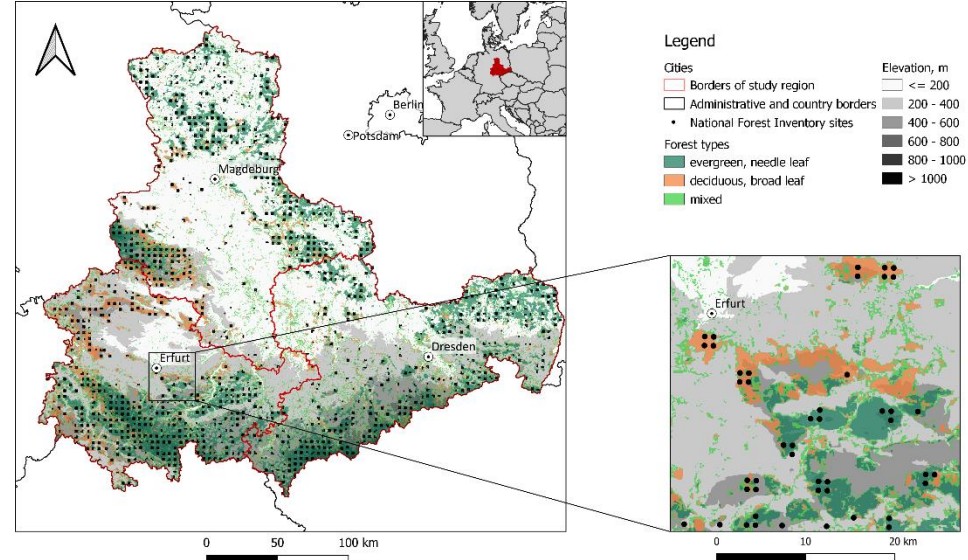

**Figure 1. Overview of the study region and selected National Forest Inventory forest monitoring sites.**

                **Table 1. Summary on the dominant tree species from the selected NFI sites**

|  | Saxony | Saxony-Anhalt | Thuringia |
|---|---|---|---|
| Spruce | 492 | 148 | 569 |
| Pine | 263 | 614 | 154 |
| Oak | 65 | 121 | 39 |
| Beech | 38 | 105 | 287 |
| Other tree | 74 | 158 | 79 |
| *Total* | *932* | *1146* | *1128* |

### 2.2.3 Local soil form map for the forested areas

State authorities of SN (Forestry of Saxony or Sachsenforst in German), SA (State Forestry Centre of Saxony-Anhalt or Landeszentrum Wald Sachsen-Anhalt in German) and TN (Forestry of Thuringia or Thüringerforst in German) provided us with the outcome of a long-term soil field campaign which resulted in high resolution local soil form ('Lokalbodenformen' in

German) maps for the forests (Petzold et al., 2016). The local soil form is a subdivision of the (main) soil type that takes into account finer differences in the substrate, the deposition features, and the horizon arrangement. Additional criteria for their identification can include nutrient and humus ratios, which have significance for tree species selection and soil treatment. To distinguish these soil forms, local place names are typically added in front of the main soil type, for example, for the main soil type 'Gneiss Brown Soil', local form 'Ölsengrunder Gneiss Brown Soil' could be identified. This allows creating a high-

resolution soil mapping (1:10 000, where the smallest area-mapping unit is about 0.5 ha), This dataset is based on the soil profile excavation data as well as NFI data and covers almost all the forested area in the study region. The joint map from three states consists of approximately 363000 polygons, containing more than 2500 unique local soil forms. For the technical reasons (interactive web-representation of the results), shape-based map was converted to point-based dataset using geometric centroid

function (nevertheless ensuring that point falls within polygon boundaries). Finally, for each node, the dominant forest type

was identified based new high-resolution (10 m) 'Dominant tree species for Germany' dataset (Blickensdörfer et al., 2022).2.2.3 Forest climate stations

Another source of the monitoring data are the local forest authorities. Currently, Forestry of Saxony provides access to the operational information from 21 forest climate stations ('Waldklimastationen' in German).All stations are located inside or directly on the edges of the forest areas (see section 2.2.3 and Appendix A1). Provided data includes not only standard

meteorological parameters, but also soil moisture under grass vegetation at 30 cm depth of a daily time scale. These grass soil moisture measurements are also of importance to the foresters, since they help to predict the changes in moisture conditions if the forest stands before they occurs, since typically grass vegetation has a faster reaction to the changing meteorological conditions than forests. The REST-API (Application Programming Interface that conforms to design principles of the REpresentational State Transfer architectural style) access to the data is provided via 'OpenSensorHub' (https://sh-

rekis.hydro.tu-dresden.de/) under cooperation with Pykobytes GmbH.

## 2.2.4 Soil moisture measurements

Soil moisture observations (soil water content using Time-Domain Reflectometry method) used for the framework evaluation were collected from Forestry of Saxony, Forestry of Thuringia and Integrated Carbon Observation System (ICOS) portal (Bernhofer et al., 2024a, b, c; Knohl et al., 2024; Rebmann et al., 2024) and FLUXNET portal (Pastorello et al., 2020) (Figure

2, Appendix A1). In total 51 stations and 245 sensors with daily time-series covered all studied vegetation types, as well as different geographical conditions in the region (except for Saxony-Anhalt, where data from only one station were found). Elevation of the stations ranges from approximately 100 m in lowlands (northern parts of Saxony) and up to 850 m in Ore Mountains. Mean annual precipitation sums (1991-2020) vary from 600 mm in northern region up to 1200 mm for the elevated southern parts. From the total number, 22 stations are located under grass vegetation (primarily in Saxony), 10 under spruce,

3 under pine, 6 under oak and 9 under beech (primarily in Thuringia) forests. Most common soil types include variations of brown soils, gleysols and podzols. All stations (except for forest climate stations under grass in Saxony) conduct measurements at multiply soil horizons (within a range from 5 cm up to 100 cm); some of stations have multiply plots (up to 6). Time-series length for each station, as well as for each horizon and sensor is highly variable. The majority of the time-series have a length 2-8 years (5.2 in median) and some go up to 25 years.

Raw time-series were superficially checked for doubtful or low-quality values. We did not perform in-deep analysis and advanced corrections or filtering. In a very few cases, data from several sensors was removed manually after personal communication from data providers. These cases include i.e. clear outliers, soil moisture sensor change/movement, animal/tree root disruptions.

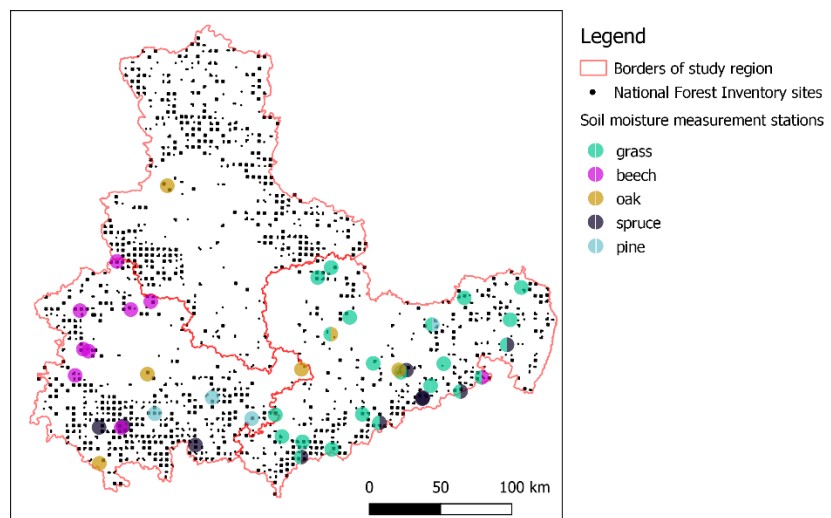

**Figure 2. Overview of the soil moisture measurement stations (see also Appendix A1). Note that for some stations in Saxony, dominating vegetation symbol is splitted in two due to the map scale (same station with grass and forest measurements)**

## 2.3 Framework description

### 2.3.1 1D physically-based water balance model (LWF-BROOK90) and its parameterisation

The LWF-BROOK90 water balance model (Hammel and Kennel, 2001) and focuses on detailed atmosphere-plant-soil water exchange on a 1D scale and is well known for its accurate representation of evaporation and vertical soil water movement processes. 'LWF' acronym states for Bayerische 'Landesanstalt für Wald und Forstwirtschaft' (The Bavarian State Institute of Forestry), where the original model was first modified. The model a branch of the BROOK90 model originally developed by Federer et al., 2003 and recently adopted in R-package 'LWFBrook90R' (Schmidt-Walter et al., 2020). As its ancestor, LWF-BROOK90 The model applies modification of the Penman-Monteith approach by Shuttleworth and Wallace (Shuttleworth and Wallace, 1985) which allows separate evaporation process into its components: interception, transpiration, and bare soil (or snow) evaporation. This approach uses a 'single big leaf' concept and two layers, separating canopy and soil. The soil profile could be represented with multiple layers. The water movement within the soil column could be divided into matrix and macropore bypass flow. It is controlled by matrix potential as well as potential evaporation and is described with the Darcy-Richards equations using the Mualem-van Genuchten hydraulic parameterisation (van Genuchten, 1980), instead of Clapp and Hornberger (Clapp and Hornberger, 1978) used in BROOK90. Additionally, LWF-BROOK90 is capable of including dynamic temperature-based vegetation characteristics (i.e. bud-burst and leaffall timings, leaf-area-index variations). For the study, a set of land cover parameters was created to represent beech, oak, spruce and pine mature forests commonly widespread in the region (Appendix A3). Additionally, grassland land cover was considered as a reference. This parameter set represents site topographical conditions, plant stand with its stem, leaf coverage and roots parameters. The set is based on the standards for temperate deciduous broadleaf and evergreen coniferous forests and grasslands proposed by Federer et al., 1996 with adaptation to the site-specific conditions of Middle Germany. Site aspect and slope were calculated from SRTM30 (NASA

JPL, 2019) digital elevation model. Forest height was set according to the GEDI dataset (Potapov et al., 2021). Main plant parameters like reflectivity, conductivity and stomatal behaviour, roughness, interception, root parameters were assigned based on existing extensive reports on the model application for the forests in Germany (Weis et al., 2023; Wellbrock et al., 2016), Level II data from forest monitoring plots (https://bwi.info/Download/) and measured and calibrated BROOK90 parameters for FLUXNET towers in Saxony (Vorobevskii et al., 2022). The vegetation phases are determined dynamically using 'vegperiod' package (Orlowsky et al., 2008), which incorporates several methods for seasonal variations of foliage growth. Budburst timing is based on method of Menzel, 1997, applicable for various tree species. For that, the cumulative heat sum (starting from February of current year) is estimated from daily temperature and tree-specific 'heat' threshold (3-6°C). Then critical budburst temperature is calculated from number of chill days (starting from November of the previous year) using another tree-specific 'chill' threshold (7-9°C). Budburst day is assigned to the day, when heat sum exceeds estimated critical temperature. For the leaffall phase von Wilpert, 1990 method was used. It is based on the 7-day moving averages of daily mean temperature and assigns leaffall to a day, where at least five previous days had temperatures under 10°C. Additionally, it 'restarts' vegetation period, if more than five consecutive days moving average temperatures are over 10°C. Finally, if no cold period fulfilling the first criteria was found, a default leaffall date is assigned to 5 October. Seasonal course of Leaf Area Index (LAI) is then determined from maximum annual LAI, budburst and leaffall dates and duration using integrated 'b90' model. It uses budburst and leaffall dates and duration, maximum (which assumed to be constant for the vegetation period) and winter LAI values (as fraction of maximum value) to create daily time-series by simple linear model. Root distribution over the soil profile is controlled by 'betamodel', which calculates relative root length densities from the cumulative proportion of roots derived by model after(Gale and Grigal, 1987). For that it uses maximum length, depth of roots and curve shape parameters (Appendix A3), which were assigned after Weis et al., 2023. It should be pointed out that long-term vegetation changes (i.e. tree ageing) are not considered in the presented operational-mode framework.

Soil parameters for the model were taken from the NFIWADS database, specifically created for NFI plots (Schmidt-Walter et al., 2019). Physical soil properties (layers, depth, texture, bulk density and coarse fragments) were obtained from the NFI soil profile database (Petzold and Benning, 2017). This database contains the predominant soil types for the inventory plots, based on the area within a 20 m radius of the centre of a plot. These soil types are linked to detailed soil profiles compiled by federal soil experts and based on the best available data in federal soil information systems for the NFI plots. Typically, all soil profiles contain information to a depth of at least 2m, unless the subsoil (bedrock) appears before. Based on this data, the hydraulic properties for the mineral soils, which are needed for Mualem-van Genuchten parameterisation of water retention curve and conductivity functions (van Genuchten, 1980) were determined using Wessolek pedotransfer function (Wessolek et al., 2009). The shape of the water retention curve thereafter could be determined using following equation (Eq. 1):

$$\Theta_\Psi = \Theta_R + \frac{\Theta_S - \Theta_R}{[1+(\alpha|\Psi|)^n]^{1-\frac{1}{n}}} \qquad (1)$$

where $\Theta_\Psi$. $\Theta_R$, and $\Theta_S$ are volumetric soil moisture contents for current, residual and saturated matrix potentials respectively, $n$ parameter is a measure of pore-size distribution, $\alpha$ parameter related to inverse of the air-entry, saturation conductivity and tortuosity parameter related to pore connectivity. The unsaturated water conductivity could be determined as follows (Eq. 2):

$$K_\Psi = K_S \frac{[1-(\alpha|\Psi|)^{n-1}(1+(\alpha|\Psi|)^n)^{-1+\frac{1}{n}}]^2}{[1+(\alpha|\Psi|)^n]^{\left(1-\frac{1}{n}\right)x}} \qquad (2)$$

where $K_\Psi$ and $K_S$ are the hydraulic conductivities for current and saturated matrix potentials respectively and $x$ is tortuosity parameter.

Due to absence of data about forest floor horizons, a uniform root-free floor horizon (6 cm) was added to profiles to provide uniform soil evaporation conditions. For the organic soil horizons, retention and conductivity parameters were taken from Wösten et al., 1999. The depth of the soil profile was organised by dividing it into layers of increasing thickness.

### 2.3.2 Meteorological data

Meteorological data needed to force the LFW-BROOK90 model is collected from open-access database – German Meteorological Service (Deutscher Wetterdienst - DWD) File Transfer Protocol (FTP) server (https://opendata.dwd.de/climate_environment/CDC/observations_germany/climate/daily/). The following variables are used on a daily scale: minimum, maximum and mean air temperature, precipitation, wind speed, relative humidity and sunshine duration. At first, the summary table on all available meteorological stations in Germany provided and regularly updated by DWD ('Stationlsexikon') was filtered and reduced based on spatial (study region plus 30 km buffer zone) and temporal (data availability starting from 2010) principles. In total 147 climate and 619 precipitation stations matched the criteria (Appendix A2). According to FTP metadata, meteorological data for the last day is added to the files between 9 and 10 a.m. of the current day. However, our tests revealed that usually real updates inside the files happen one or two hours later. Furthermore, station data is splitted in parts: 'historical' and 'recent' (data from the current year).

In reality, available meteorological data which could be found and retrieved from the DWD FTP server significantly vary from the selected station short-list (Fig. 3). Up-to-date (1-day lag) air temperature and relative humidity data is available from around 115 stations, sunshine duration and wind data could be retrieved from 31 and 54 stations respectively. Finally, precipitation measurements are represented with 330 up-to-date stations. Still, the spatial coverage of meteorological data for all the variables in the study area can be considered good and homogeneous. Although the exact number of available stations with up-to-date data vary from day to day, tests of the meteorological data retrieval for 10-year simulations during the June-October 2023 period did not show large deviations in station data availability (< 5% of total station number).

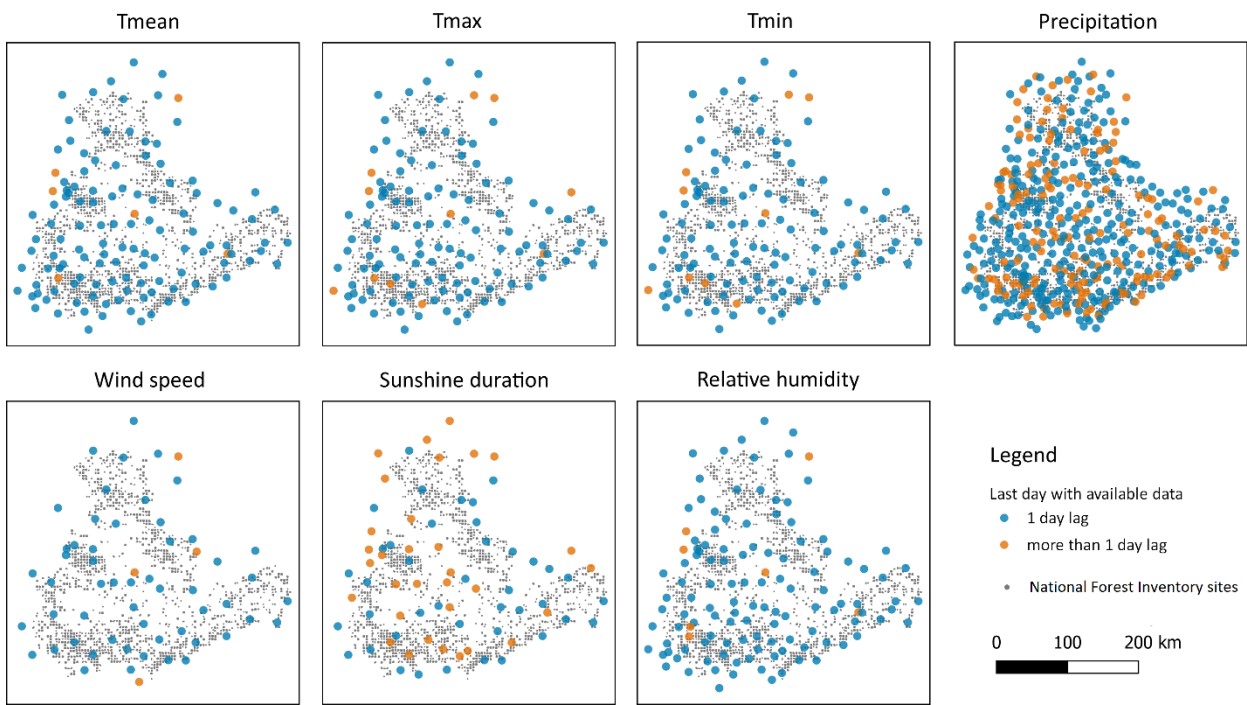

255

**Figure 3. Real-time meteorological station data availability for different variables for the study area on a daily scale: last date with not missing data (retrieved from File Transfer Protocol server of German Meteorological Service at 12:00 20.09.2023).**

Figure 4 summarises the availability of actual (1-day-lag) meteorological data with regard to the distances to the NFI plots. Median distance to the nearest station with temperature and relative humidity is about 11 km, while for the less dense wind

260 and radiation measurements the closest station is on average 15 km away. Precipitation stations due to higher density are typically found in much shorter distances of approximately 5 km. These from the first sight relatively high distances do not necessarily dictate the output resolution of the presented framework due to the several reasons. At first, it should be taken into account, that variables such as temperature and solar radiation do not typically possess large spatial heterogeneity within the resulted average distances, which could bring a significant influence on the modelling results. Further, for each simulated point

265 within the same NFI plot, meteorological input will be different due to different distances to stations, even if the filtered meteorological stations are the same.

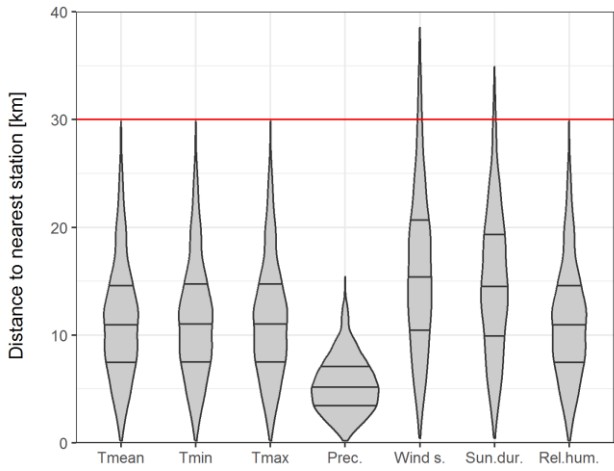

**Figure 4. Distances to the nearest meteorological stations with available 1-day-lag data for National Forest Inventory plots. Red line represents the chosen 30 km buffer zone. Horizontal lines inside respective violin plots represent the 25th, 50th and 75th quartiles.**

For each NFI site, the meteorological input data to force the model is prepared in the following way. For all the necessary variables separately, framework scans for the nearest stations with available data within the 30 km buffer zone (Fig. 5). Filtered stations are thereafter checked for a real data availability. Stations with more than 5% of missing data (within simulation time period) are not further considered. Afterwards, inverse distance weighted mean is applied to create one time-series from filtered stations' data. In case there was no station found within the default buffer zone (e.g. some wind speed and sun duration data for some of the plots) or, 1-day-lag data is not presented in any of the filtered stations, the framework expands its buffer to take the next nearest station, which matches data criteria. To avoid too much smoothing of local weather patterns in case of too many stations available around the point, which is especially important for precipitation, the maximum number of stations, which are considered, is limited to seven nearest ones. Temperature and wind speed data are thereafter used directly as model input. Sunshine duration is converted to global radiation using site latitude and day of year according to Angstrom equation (Angstrom, 1924). Vapour pressure deficit is calculated from relative humidity and mean air temperature using Magnus equation (Alduchov and Eskridge, 1996). Precipitation values are corrected for the systematic measurement error (i.e. drifting, wetting, evaporation) using mean air temperature, day of the year and 'well-protected' station-shielding type using Richter equation (Richter, 1995). Finally, the framework checks if any NA values appear in the prepared time-series. So far, we did not experience such a case, since typically the procedure mentioned above should preserve all possible NA appearance. Nevertheless, if NA values are still found in the prepared data (i.e. can happen for wind or sunshine duration data, as the station network is sparse; or there is an update failure by certain DWD stations), they are filled with monthly-mean values.

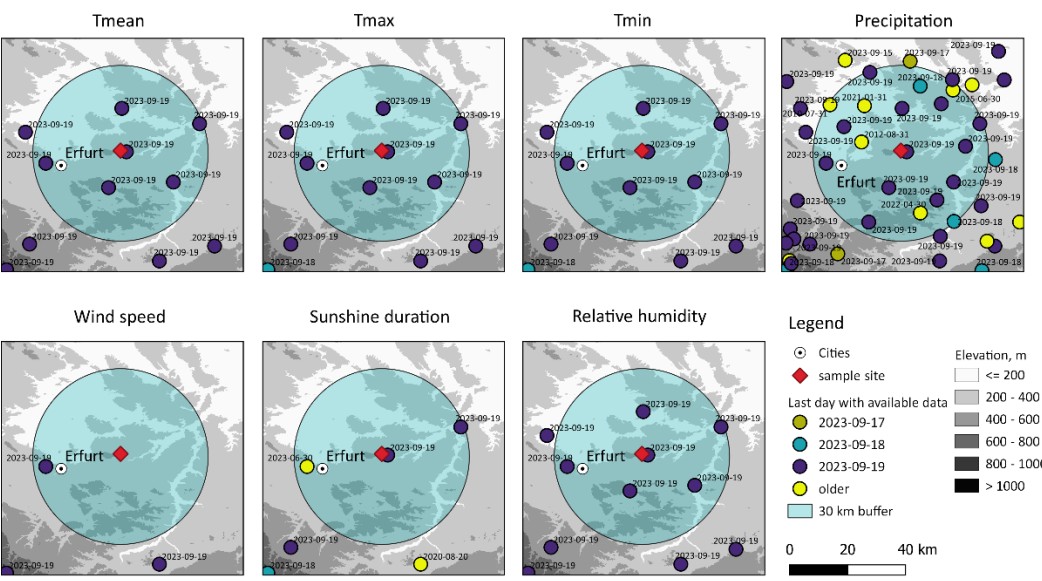

**Figure 5. Subset of available meteorological data for a selected National Forest Inventory site (retrieved from File Transfer Protocol server of German Meteorological Service server at 12:00 20.09.2023).**

LWF-BROOK90 uses precipitation input on a daily scale as default and disaggregates the data into hourly resolution internally using the 'pdur' parameter (average duration of daily precipitation event in hours for each month, 4h for each by default). Therefore, to improve correct separation between canopy interception, overland flow and infiltration, it was decided to estimate this parameter more accurately based on real data from meteorological stations in the area. For that, three meteorological stations located in all three study area states with hourly precipitation time series (1995-2022) were picked: Erfurt-Weimar (ID#1270), Magdeburg (ID#3126) and Dresden-Klotzsche (ID#1048). Resulting monthly duration values averaged over three stations were found to be 3h in all months except for November (4h).

### 2.3.3 Operational mode and website

The first pilot version of 'Soil Moisture Traffic Light' was released in 2022 (Kronenberg et al., 2022). The setup was successfully validated using soil moisture measurements from eight grassland and three forest sites for the 2006-2019 time-period (Luong et al., 2023). The novel framework, initially developed and tested for Saxony, has was upgraded in its architecture, improved by handling of the meteorological input and model parameterization and has been expanded spatially in this paper.

The presented upgraded framework for three German regions is functioning in an operational automatic mode with 24h update time (Fig.6). It starts each day at 11 a.m. with the download and pre-processing of 1-day lag meteorological data, which lasts about 3 min. Afterwards, the simulations for each NFI site (with four dominant tree types and grassland, resulting in 3206 plots * 5 vegetation types model runs) and forest climate stations (with grassland for 21 stations) are run for a 10-year period using parallel processing, which takes approximately 40 min with 48 cores and could be potentially reduced by applying more

cores. Despite the fact that the tree-species composition at each NFI site is known, in some of the cases it is rather a mixed

forest with no clear predominant tree type, moreover, forest managers requested that the results should be presented for each of the selected tree species at each plot. For the local soil types, the results were assigned from the neighbouring NFI plots based on dominating forest and soil type. Raw model runs are post-processed and stored in 'csv' and 'geojson' files, which can be downloaded from the website. Additionally, model was run for a longer rime period (01.1990-04.2024). Results were used for the framework validation, analysis of water balance components and determination of soil water content quantiles.

Soil moisture conditions are represented with the Plant-Available Water (PAW) coefficient (Eq. 3), which is calculated for each presented soil layer and indicates the amount of water left in the soil accessible to the plant before it starts to wilt:

$$PAW = \frac{\Theta_\Psi - \Theta_{wp}}{\Theta_{fc} - \Theta_{wp}} \qquad (3)$$

where $\Theta_\Psi$. $\Theta_{wp}$, and $\Theta_{fc}$ are volumetric soil moisture contents for current, wilting point and field capacity matrix potentials respectively. Here, the soil moisture at wilting point and field capacity is derived from Eq. 1 using pressure head of -1585 kPa

and -6.3 kPa respectively (Schmidt-Walter et al., 2019; Wessolek et al., 2009). Values of PAW greater than 1 indicate oversaturation (soil moisture is higher than field capacity) and values lower than 0 represent situation, when soil water content is below the plant-extractable threshold (i.e. water remains only in smallest pores and as films or is chemically-bonded to soil particles).

The final product of the 'Soil Moisture Traffic Light' can be assessed online via https://life.hydro.tu-

dresden.de/BoFeAm/dist_bfa_kk/index.html. It was developed using the Node Package Manager in conjunction with the Javascript runtime environment Node.js using OpenLayers and ApexCharts Java libraries. These enable the web display of geodata and various data visualisations. As the main stakeholders are German authorities, the website' main version is in German, however, to make the product more attractive and usable for the international community, an English version is available as well. The framework uses two virtual machines provided and managed by the Center for information services and

high performance computing (ZIH) of TU Dresden.

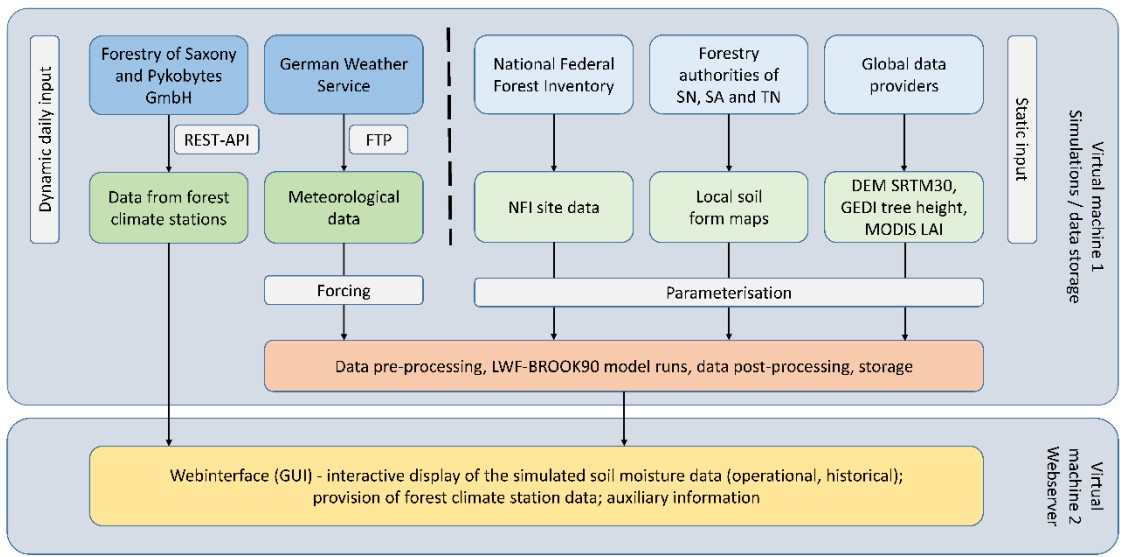

**Figure 6. Framework architecture.**

## 2.4 Framework validation scheme

One of the most important part in the validation is the selection of an appropriate NFI site from the framework for each
measurement station, since the system is not designed to simulate these exact locations including all local specifics
(topography, soil properties, and vegetation). At first all available information on the tree composition from each station (which
amount and quality is various for different sites) was summarized and upscaled to a dominant specie (i.e. all variations of
beech trees were reduced to 'beech' type), resulting in the same number of vegetation classes as for simulated NFI sites. Then,
local soil form for each station was derived from the available map (see section 2.2.3), since original station metadata does not
include this information. Afterwards, for each station a nearest corresponding NFI site was selected based on the match of
local soil form and soil type in the 10 km radius (Fig. 7, left side). For majority of stations, respective NFI site was found
within a 1-2 km radius.

Further, for each soil moisture sensor corresponding layers from associated NFI soil profile were selected. Various possible
combinations were constructed to account for possible uncertainties due to sensor positioning relative to available soil layers
and vertical footprint of the measurements (Fig. 7, right side). Here, for example, sensor 2 with 50 cm depth was compared to
6 time-series from the simulated soil moisture from layers 2, 3 and 4 and their weighted mean (by layer thickness)
combinations.

As metric to compare observed and simulated soil moisture, the Pearson correlation coefficient on a daily scale was considered.
Additionally, absolute the bias values and Kling-Gupta-Efficiency (KGE) were evaluated. To aggregate the results from
350  different sensors, plots and stations for the various domains (e.g. same sensor depths, station, vegetation, or whole framework)
a weighted mean (by data length) was calculated due to very heterogeneous lengths of available time-series.

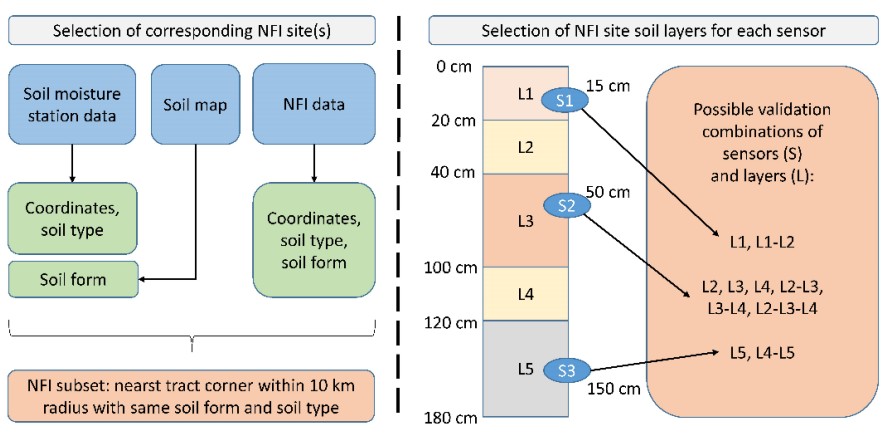

**Figure 7. Selection of corresponding NFI site(s) and soil layer(s) for each soil moisture sensor**

# 3 Results

## 3.1 Comparison of measured and simulated soil water content for the forest and grass sites

The framework performance for all 51 stations is presented in Fig. 8. Validation results for each station and sensor could be found in Supplementary (Vorobevskii, 2024). Here the results are summarized for three soil depth categories (0-20 cm, 20-50 cm and 50-100 cm), two different time periods (April-October as growing and November-March as winter seasons), five vegetation classes and three classes of available data length (1-5 years, 5-10 years, and 10-25 years). The validation showed a discernible seasonal variation in the correlation coefficients, with higher values observed during the vegetation period (0.76 in median) and lower values during the winter period (0.71 in median). The reduced correlations during the winter months can be attributed to increased uncertainties in both simulations and observations, especially for frozen soils and snow cover, as the quality of the sensors might be compromised, as well as the lack of soil frost accounting in the model. Further, the performance of the framework has significant variations among the presented vegetation types. It was found better for deciduous (0.81 and 0.86 in median for beech and oak respectively), rather than for coniferous (0.74 and 0.64 in median for pine and spruce respectively) forests. The results suggest a need to re-evaluate the spruce parameters, since the systematically worse performance was detected for most of the spruce stations and sensor depths. The bandwidth (space between $25^{th}$ and $75^{th}$ quantiles) of obtained correlations in general increases with the soil depth. With regard to the profile depth, validation showed a better agreement (0.83 in median) of modelled and observed soil moisture in the upper horizon (0-20 cm), while in lower horizons (20-50 and 50-100 cm) model performance was found lower (0.74 and 0.64 in median respectively). Results for different plots and sensors aggregated by station have shown that the obtained median correlation coefficients variate more for the grass (0.31-0.86) and spruce (0.41-0.82) vegetation types, while the lowest variation was found for beech stations (0.71-0.83). The best and the worst performance was found for several grass stations in Saxony: 0.31 in Olbernhau and 0.86 in Graupa. Best performance for the forest stations were found for beech and oak stands (e.g. 0.83 in Harz and Hohes Holz).

The analysis of the obtained Pearson correlation coefficients with regard to site characteristics (Appendix A4.2) did not reveal consistent trends for sensor depth, soil type, soil form or distance to the NFI plot used for the evaluation. However, it was found, that model performance was slightly worse in the mountainous regions, where site elevation was higher than 600 m and annual precipitation was over 1000 mm. Furthermore, we discovered that longer data records have a positive feedback on the model fit. Finally, the framework performance was found noticeably better for the stations in Thuringia, rather than in Saxony.

Evaluation of the KGE values (Appendix A4.1) showed, that for the majority of the stations (except for four spruce and three grass sites) the values lie above the critical value of -0.41 (Knoben et al., 2019), thus the modelled soil moisture is better than mean of observed one. In general, low KGE by soil moisture evaluation is associated with bias and variance ratio errors (up to 200%), which were found systematically overestimated in the topsoil and underestimated in the lowest horizons. Here, highest KGE values for the forests were found for the topsoil horizon under pine stand (0.56), while the lowest were shown by spruce

stand for the 50-100 cm depth (-0.57).

Overall, the obtained validation results are fairly consistent with previous studies by Boeing et al., 2022 and Xia et al., 2014 who extensively validated soil moisture simulations from different hydrological models used for drought monitoring in Germany and USA. However, it should be noticed, that, in comparison to the other studies we evaluated raw time-series, which did not undergo any post-processing (e.g. ranked, deseasonalized, anomaly-corrected). There transformations, which are

typically advised for the validation of grid-based soil moisture simulations with point measurements (Crow et al., 2012), were not applied in the study intentionally, in order to assess and demonstrate the extent of the ability of point-based framework to explain the original and non-modified variability of soil moisture.

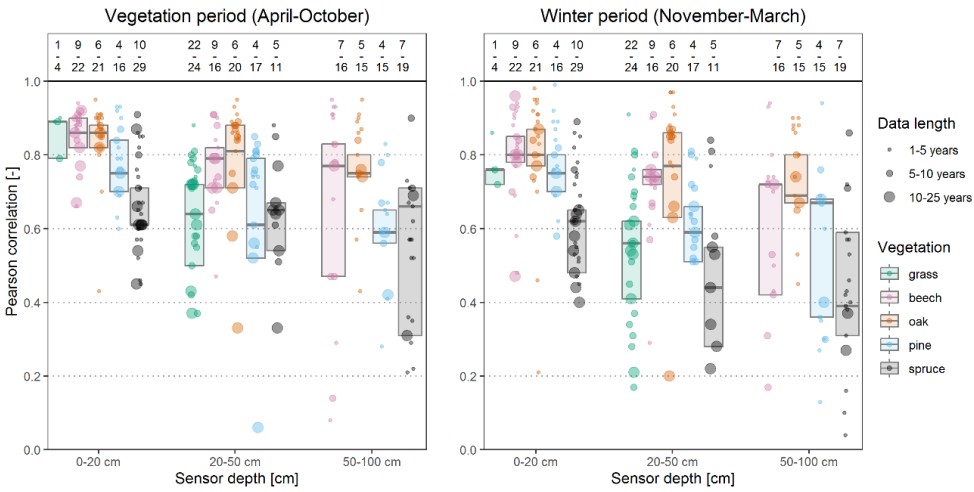

**Figure 8. Pearson correlation between simulated and observed soil moisture for all stations and sensors (daily scale). Boxplots**
**without whiskers represent 25-75% quantile range and median calculated from weighted values (after data length). Numbers above state for the number of stations (numerator) and sensors (denominator).**

**3.2 Effect of the local scale on the soil moisture under drought conditions**

To illustrate the advantages of the point-based framework two examples were chosen. Local-scale effects of precipitation front which occurred after a dry week (no recorded precipitation for almost all meteorological stations) over the study region on the soil moisture is shown on Figure 9. It could be seen (Fig. 9a), that the daily precipitation sums on the autumn day of 12.09.2023 decrease from 20-30 mm on the north-west (some stations recorded up to 48 mm) to 0.1-1 mm on the south-east (about 60 stations mostly located in Ore mountains did not record any precipitation on that day). The soil moisture situation under spruce forests for the major parts of the region before the rain event could be expressed as a drought condition, as the precipitation sums for the August and beginning of September were below the climate means while the potential evaporation was still high. PAW values (Fig. 9b) in the lowlands were mostly below 0.1, while in the mountainous regions enough plant-available water (PAW>0.6) was simulated, which correlates well with the topography and soil textures distribution. This soil moisture situation is also confirmed by absolute values of volumetric water content (Fig. 9c). Thus, low PAW values correspond to 4-15% of water content, while normal and wet conditions (PAW>0.4) showed volumetric soil moisture content in general between 15% and 30%. As expected, under the conditions of severely depleted water storage in the soils in lowlands, even a considerable amount of precipitation did not result in significant improvement of drought conditions for the plants or soil water storage recharge. NFI sites located near the meteorological stations, which registered more than 10 mm showed on average 0.1-0.15 PAW (or 4-6% for volumetric moisture) increase, while for the rest, the change was on average less than 0.05. Overall, all the local changes for 12.09.2023 followed the precipitation patterns. By comparison of the two selected depths, it could be concluded that the upper soil layers (up to 40 cm) showed more distinct reaction to the rain event. The lower horizons (up to 100 cm), which on 11.09.2023 had lower mean soil moisture content (especially for sites with PAW>0.3), were almost not affected by the event. Here volumetric soil moisture increased on average only by 1-2 % even in north-western parts. This could be explained by the fact, that under general drought conditions, infiltration of even a substantial rainfall amounts into deeper horizons was prevented in the topsoil by root uptake and thereafter transpiration process.

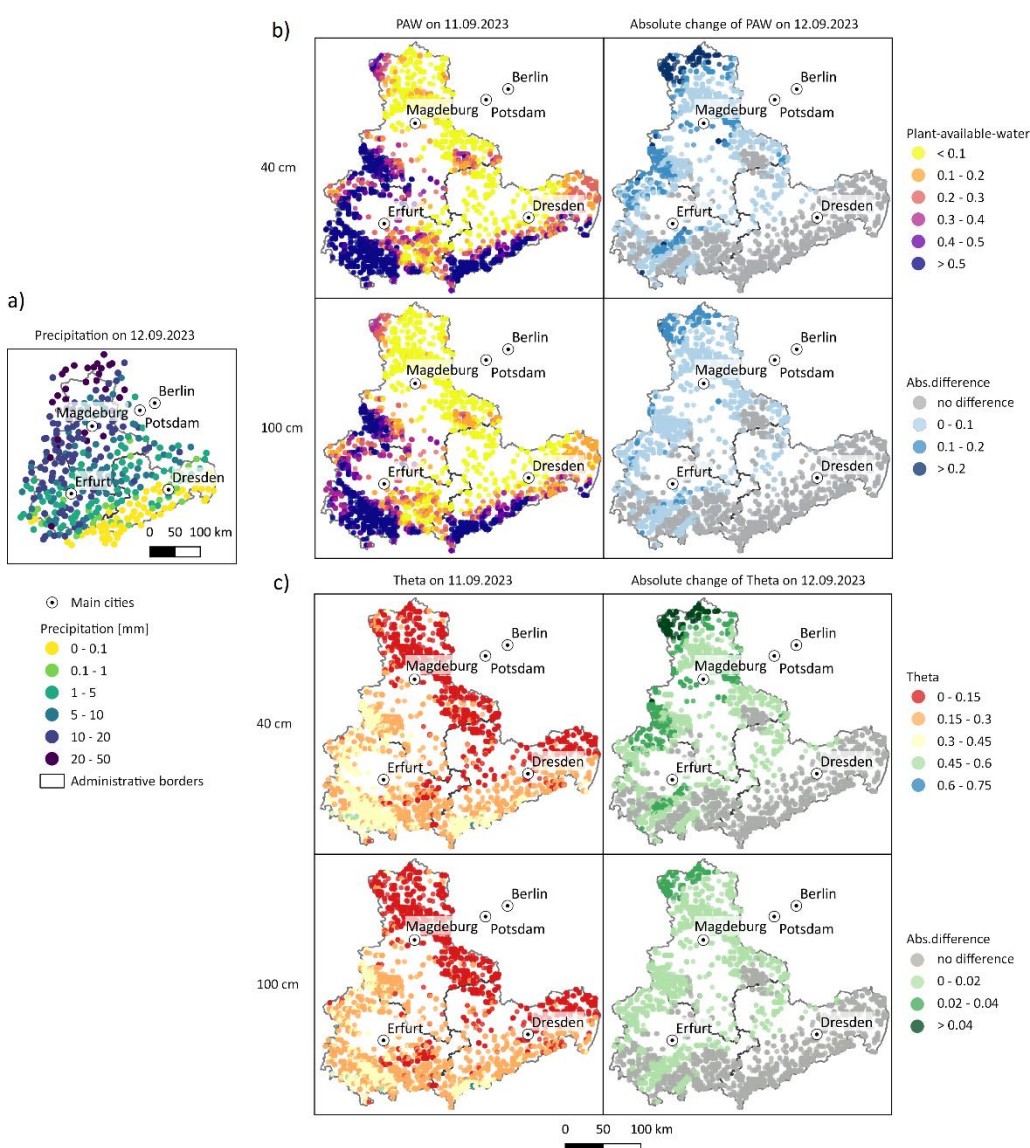

**Figure 9. Spatial effect of one-day rainfall (a) after a dry week on plant-available water (b) and volumetric soil moisture content (c) in spruce forests for 40 and 100 cm profile depth**

The effect of the local-scale topography on the depletion of the soil moisture storage in the beginning of the 2018-2019 drought period is shown on Figure 10. Here both slope and aspect influence the net radiation and downslope flow, thus affecting the potential evaporation and soil column outflow rate respectively. Thus, south-orientated site (#56712_3) with an almost 25-degree slope (Fig. 10a) showed up to 20% lower PAW values than north or west sites with flat slopes, especially in the drought propagation phase (Fig. 10b). Moreover, different tree species exhibited varying reactions and sensitivities to changes in soil moisture based on the topographical features. During the growing season (spring 2018), this impact was notably pronounced in deciduous forests owing to the higher available soil water resulting from reduced transpiration and interception rates.

However, during summer period, the influence of slope and aspect became more significant in spruce forests due to elevated total evaporation rates (which are indicated by larger leaf area and stem area indexes compared to beech forests).

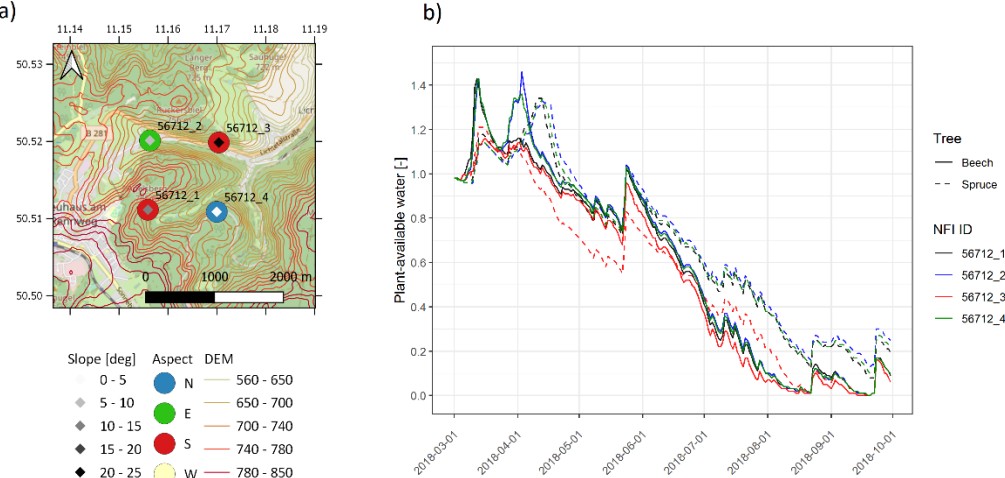

**Figure 10. Effect of slope and aspect (a) of NFI plot (#56712, near Apelsberg, Thuringia) on the simulated plant-available water values (up to 100 cm depth) in spruce and beech forests for the drought propagation phase in 2018 (b).**

### 3.3 Accounting for seasonality in soil moisture changes during vegetation period

As the study region has heterogeneous topographical, climatological and landscape features, they all have a strong influence on the seasonal course of soil moisture. As an example, fortnight changes of PAW up to 1 m depth under beech forests for a 'normal' (climatologically, meaning not abnormally dry or wet) year of 2020 are presented (Fig. 11). In January, soil moisture in the central and northern parts is still not recovered from the dry season of 2019, especially in the eastern parts of Saxony, where the PAW values were around 0.2. Starting from February until the end of March, precipitation amounts and snowmelt combined with low evaporation rates (deciduous forest, thus mainly only stem interception and soil/snow evaporation occur) refill soil with moisture until its maximum annual values. High-elevated areas in the south of Thuringia and Saxony, as well as western and northern parts of Saxony-Anhalt received therefore oversaturated soils (PAW >1.2) for more than a month, while middle part of the area still remained unsaturated (PAW<0.6). Thereafter in the beginning of the vegetation period (growing phase in April-May) soil moisture mostly remained in normal stable conditions (0.6< PAW<1), while constantly, but slowly increasing evaporation values were still compensated by relatively high precipitation input. Starting from June and till the end of September transpiration clearly dominated among evaporation processes and coupled with typically lower rainfall sums, let the soil water storage quickly deplete and remain almost in an empty state (PAW <0.2). Thus in this time period, almost all the precipitation input was immediately consumed by the forest and did not contribute to refill of the soil profile. Situation started to straighten out in the middle of October with the leaffall and thus a reduction of total evaporation, so that the fallen rainfall could finally recharge depleted soil water storage. Due to typically higher precipitation amounts in the mountainous regions, this process started and was more clearly visible exactly in the southern parts of the study area, while in

the middle and northern typically flat and dryer parts, the moisture refill was much slower and less effective. Thus, by the end December, the spatial distribution of the soil moisture under the beech forests looked similar to what it has started with in the January of 2020.

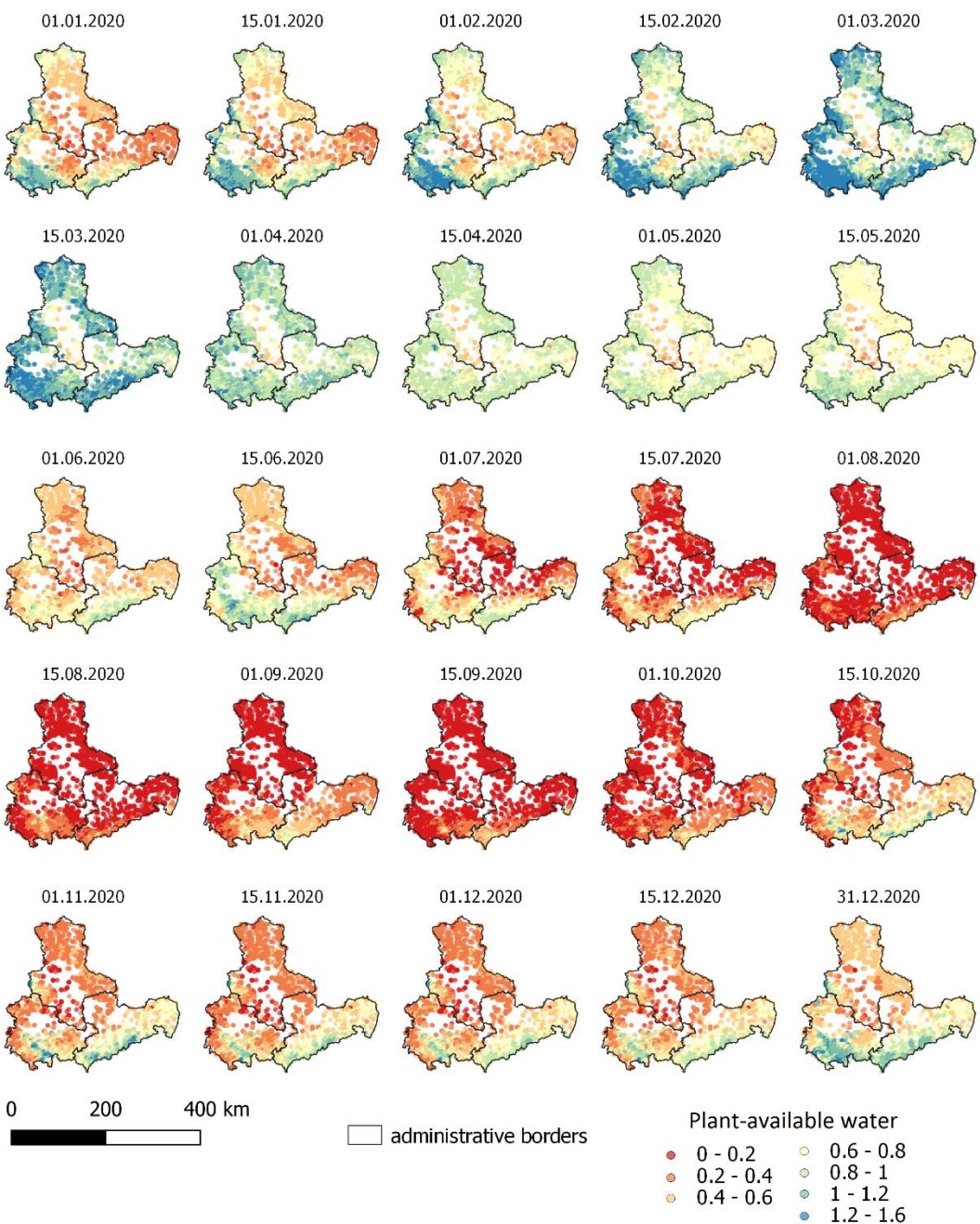

**Figure 11. Seasonal changes of plant-available water (up to 100 cm depth) under beech forests in 2020.**

455

**3.4 Influence of forest composition on the evaporation and its components**

Redistribution of precipitation input into evaporation and runoff components as well as in their sub-components is crucial for correct representation of in-situ soil moisture. Long-term simulation data (30 years, 1991-2020) for the NFI plots revealed that the evaporation-precipitation ratio (Fig. 12) in deciduous forests is lower (85% median) than in coniferous forests (95% median) and has higher range (almost 50%). Highest values were found in central and northern dry flatland areas. Evaporation of intercepted rain and snow is a substantial part of mature forests and possesses large spatial heterogeneity. It is estimated higher for spruce (350 mm/year median) and pine (330 mm/year median) forests, thereby deciduous showed much lower interception amounts (220 mm/year median). Transpiration, on the other hand, was found higher for oak (350 mm/year median) and beech (320 mm/year median) forests, while coniferous forests transpire about 270 mm/year. Soil evaporation is the smallest part of total evaporation (about 20%) and remains almost stable with regard to spatial scale and vegetation cover (80-90 mm/year median).

Although bandwidth of obtained E/P ratios for forests does cover a very wide range (0.5-1), high median E/P ratios (above 0.8) do not correspond well with other studies. For example, in the study of Renner et al., 2014, it was found, that according to watershed-based evaporation estimations (discharge subtracted from precipitation), the E/P ratio for small forest-dominated catchments in Saxony laid typically in a range of 0.3-0.7 for the 1950-2009 time-period. It was however mentioned, that actual evaporation tends to consistently increase in the last two decades. In another study by Vorobevskii et al., 2022 different BROOK90 frameworks were applied to simulate FLUXNET sites within Saxony. It was found, that the E/P ratios in forest stands based on measurements (0.50 for Tharandt, 0.41 for Oberbärenburg and 0.65 for Hetzdorf) could not be reached with standard model parameterisation without calibration, especially for coniferous forests, where modelled E/P ratios were found up to 0.8. These facts reveal the importance and sensibility of the model parameterisation, especially regarding the interception component, which should be addressed more in-depth in further studies.

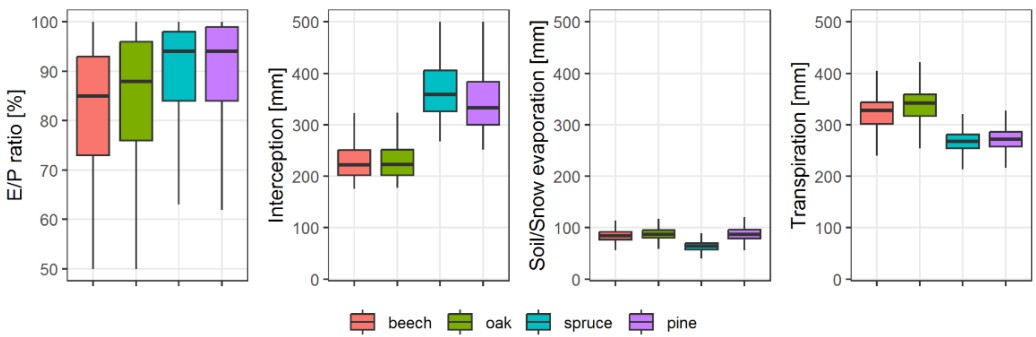

**Figure 12. Mean annual evaporation components in different forest types for 1991-2020 period.**

**3.5 Soil Moisture Traffic Light - informative web-platform for forest managers**

The website homepage (Fig. 13) shows a map with the soil moisture conditions for the forests in Thuringia, Saxony and Saxony-Anhalt. It was derived as a combination of the four tree-specific simulations conducted for each of 3206 NFI tract

corners and high-resolution soil form map. For each respective polygon (i.e. centroid) of the soil map, where identified dominant forest is one of the four simulated ones (spruce, pine, oak, beech), the framework searches for the nearest (radius is limited to 5 km) NFI site which has same soil form and assigns simulated PAW values accordingly. Additionally, simulation results for each NFI site for four tree species as well as grassland are presented separately. For easier communication of the results to non-expert stakeholders, current moisture states are divided into four colour categories: very wet (PAW>1), normal (0.4<PAW<1), dry (0.2<PAW<0.4) and very dry (PAW<0.2). Intuitive buttons allow the user to switch between different vegetation types (spruce, oak, beech, pine, grass) and different soil profile depths (0-40 cm, 0-80 cm and 0-100 cm). There is also an option to customise the background maps with Google Maps or OpenStreetMap. On the right side, the user finds supportive information of what each colour means in terms of vegetation feedback.

The expert mode offers deeper insights into the soil moisture condition of each NFI plot. By clicking on a specific point on a map (or menu bar), the user gets detailed information (Fig. 14) on the selected plot (name and ID, soil profile data, forest type, climate overview). Furthermore, on the right side two interactive graph panels appear. Upper graph (Fig. 15a) shows a temporal PAW course up to a depth of 40 cm for the current year with different quantiles and median values as a background, calculated from 30-year period historical simulations. Lower panel (Fig. 15b) gives an overview on the annual PAW developments for different soil depths up to 100 cm. In addition, expert mode provides the possibility to download raw data from each of the shown panels. Further, the simulations and observations from the forest climate stations are available (currently only for Saxony). Besides presentation of the up-to-date main climate elements, the soil moisture measurements (under grass) could be used as an indirect validation of the framework. Further, the website has FAQ and Glossary, which explains the main methods and terms behind the framework. Finally, a citizen-science approach was implemented via Feedback-survey button on the start page to continuously optimise the application.

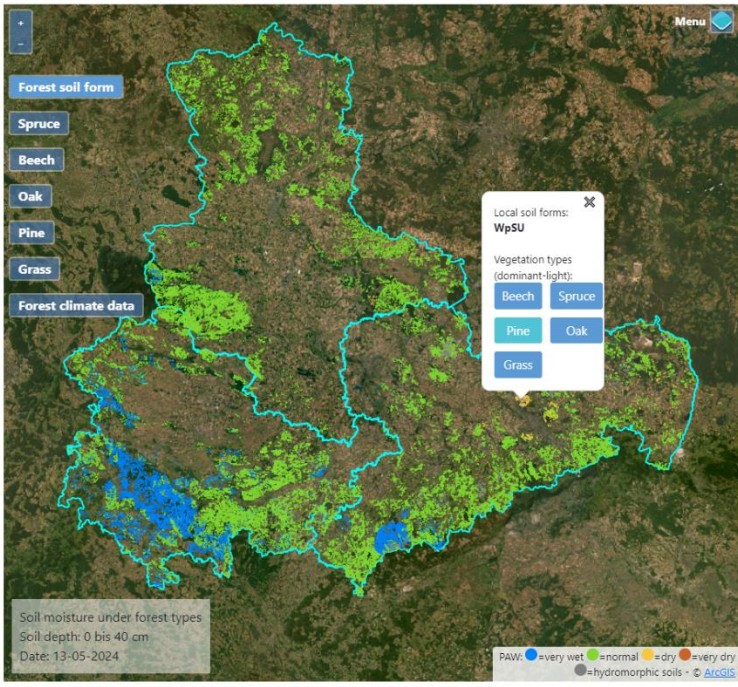

**Figure 13. Overview Soil Moisture Traffic Light main page– plant available water for combined forest types and soil form maps (13.05.2024, 40 cm depth)**

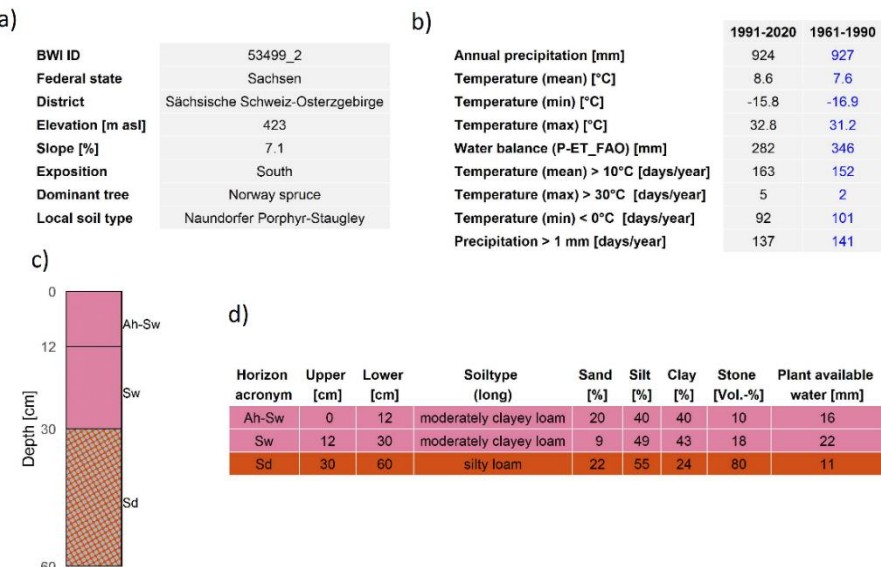

a)

| | |
|---|---|
| **BWI ID** | 53499_2 |
| **Federal state** | Sachsen |
| **District** | Sächsische Schweiz-Osterzgebirge |
| **Elevation [m asl]** | 423 |
| **Slope [%]** | 7.1 |
| **Exposition** | South |
| **Dominant tree** | Norway spruce |
| **Local soil type** | Naundorfer Porphyr-Staugley |

b)

| | 1991-2020 | 1961-1990 |
|---|---|---|
| Annual precipitation [mm] | 924 | 927 |
| Temperature (mean) [°C] | 8.6 | 7.6 |
| Temperature (min) [°C] | -15.8 | -16.9 |
| Temperature (max) [°C] | 32.8 | 31.2 |
| Water balance (P-ET_FAO) [mm] | 282 | 346 |
| Temperature (mean) > 10°C [days/year] | 163 | 152 |
| Temperature (max) > 30°C  [days/year] | 5 | 2 |
| Temperature (min) < 0°C  [days/year] | 92 | 101 |
| Precipitation > 1 mm [days/year] | 137 | 141 |

c)

d)

| Horizon acronym | Upper [cm] | Lower [cm] | Soiltype (long) | Sand [%] | Silt [%] | Clay [%] | Stone [Vol.-%] | Plant available water [mm] |
|---|---|---|---|---|---|---|---|---|
| Ah-Sw | 0 | 12 | moderately clayey loam | 20 | 40 | 40 | 10 | 16 |
| Sw | 12 | 30 | moderately clayey loam | 9 | 49 | 43 | 18 | 22 |
| Sd | 30 | 60 | silty loam | 22 | 55 | 24 | 80 | 11 |

**Figure 14. Summary information for selected NFI site (#53499_2, Tharandter Wald, Saxony): general information (a), climate data (b), soil profile (c) and soil characteristics (d)**

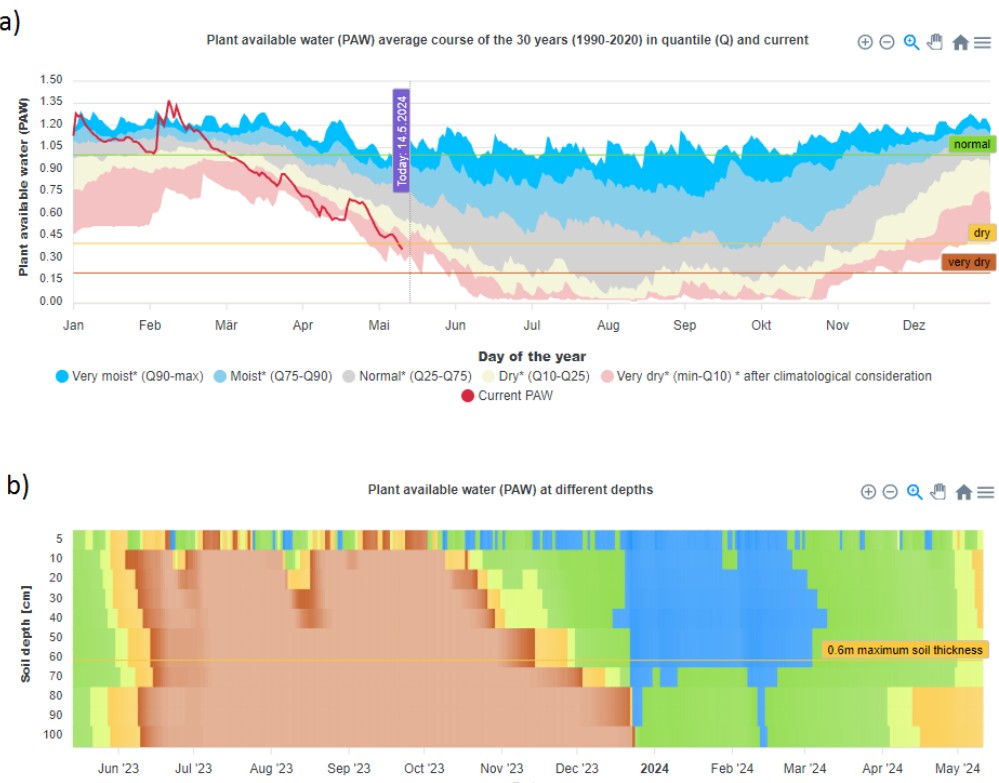

**Figure 15. Overview on the expert-mode of the Soil Moisture Traffic Light for selected NFI site (#53499_2, Tharandter Wald, Saxony) and spruce forest: topsoil (< 40 cm) plant available water for the current year in comparison to quantiles derived from the 30-year historical period (a); plant available water for all soil profile layers for the current year (b)**

## 4 Discussion

### 4.1 On the benefits and shortcomings of the developed framework and its placement among other existing systems

Soil moisture monitoring frameworks typically utilize grid or point-based data presentation mode. The choice of one upon another method relies on the system build-up: model, input datasets, and user orientation and requirements.

The main difference between grid-based and point simulations is accounting for local conditions. Raster-based setups are capable of covering the whole domain of interest without spatial gaps, however, due to high computational costs, the local scale (< 500 m) could not be reached so far (Vorobevskii et al., 2022). Thus, point-based setups could still be superior when it comes to local scale, since here differences in geographical conditions could be taken into account easier. During the development phase of the presented framework, we were in direct communication with forest managers. Their feedback showed us that the actual data availability at the point (particularly detailed soil information and its near real-time as well as retrospective hydrological behaviour) is much more of usage than the generalised grid-based information, claiming everything else as often redundant. Foresters stated that the existing systems are too coarse in resolution, which was the main intention

for the development of this point-based modelling system with point-based soil information data from National Forest Inventory and high-resolution local soil map (1:10 000), rather than generalised soil maps of coarser resolution (1:1 000 000). Raster-based simulation at these fine resolutions will require a grid of approximately 10 m, which is implausible for the implementation due to computational power requirement and technical problems of the result representation (in an interactive and not just raster-based way). From a soil hydrology perspective, it is always advantageous to map the high spatio-temporal heterogeneity of soil moisture, which is largely controlled by the local variability of physical soil properties, topography and vegetation (Kemppinen et al., 2023; Kirchen et al., 2017; Seltmann et al., 2021). Further, point modelling has the advantage that we can update point information (input parameters) and/or add new points (soil profiles) easily and at any time without having to dismantle the entire model concept. This approach therefore offers an open-space room for the continuous improvement. Point display might have practical advantages as well, since additional information from the site (soil profile and soil properties, climate summary, topography information) fits into the display, not to forget the helpful background (e.g. satellite images, topographic and soil maps). On the other hand, grid-based simulations are easily for post-processing, comparison to other similar system outputs as well as for visualisation and integration in web-tools. In comparison to fully distributed modelling, the utilized 1D LWF-BROOK90 model could not adequately represent lateral flows (especially inflow) on the sloped terrain and does not account for groundwater flow (only bucket-type retention storage, which does not contribute in any way to soil moisture). This could be critical point for the correct soil moisture estimation in the places with high slopes and elevated groundwater levels. However, according to data from the foresters, approximately 75% of forest floors (at least in Saxony) are characterised by predominant vertical top down seepage water movement and are not affected by seasonal or permanent groundwater table variations. The few NFI plots with identified potential groundwater influence (by appearance of specific soil horizons in the profile) are marked with grey colours in the website layout. Further, for correct representation of the lateral inflows and outflows (which are the undeniable advantage of grid-based models) a good-quality terrain model is desired. In Central Europe, digital elevation models (and therefore streamflow networks) are massively covered with artefacts due to the strong anthropogenic influence (bridges, forest roads, mining traces, dams, culverts, ditches) and would have to be extensively cleaned up in order to actually achieve the alleged accuracy of soil moisture modelling, which is problematic for dense vegetation like forests. Finally, accurate correction for tree height to produce high-resolution digital terrain model is possible only with laser/drone scans due to high heterogeneity of the forest stand. With regard to 'change-of-support' problem (Allocca et al., 2023; Gelfand et al., 2001) the presented system itself does not introduce contradictions into scale mismatch: both input (meteorological, soil and land cover data) and output of the framework are point-based products. However, in comparison to grid-based modelling, this solution could imply thereafter difficulties in the correct interpretation and possible extrapolation of results, as these tasks then lie on the shoulders of the end users, which in our case are mostly experts with profound knowledge of forests and local soil specifics. Therefore, our approach was to link the data available at the plot level with the best available model parameters and modelling techniques at the same spatial scale.

At the present time, two other soil monitoring systems existing in Germany, which were mentioned in the introduction, use more or less the same meteorological input (with differences in quality-control and interpolation techniques of the data).

The German Drought Monitor uses the distributed hydrological model, which accounts for lateral flow and groundwater processes, but possesses limitations with regard to evaporation representation. Original setup had a resolution of 4 km, its updated in 2022 version (Boeing et al., 2022) increased it to 0.98-1.23 x 1.7 km size (variable grid). This framework includes simplified forest classes (coniferous, deciduous and mixed) from satellite-based land cover map and soil characteristics based on a 2 km resolution map. Furthermore, soil profile depths are discretized in two layers (0-25 cm and remaining depth). For a soil moisture indicator, the soil moisture index (Zink et al., 2016) for total soil column and topsoil and plant-available water are is presented in raster (daily update) and partly as GIF formats (two-week temporal developments). In summary, the absence of additional site (in this case – grid) information, non-interactive presentation of results and general coarse resolution of the output product generated from medium-resolution soil maps make the practical application of this product in forestry difficult (Meusburger et al., 2022; Speich, 2019).

The Soil Moisture Viewer from German Meteorological Service uses LWF-BROOK90 and AMBLAV models for forests and agriculture respectively. The 'forest' part of the system was added in February 2024 and is in a test mode. The setup and parameterization of the model is similar to the one presented in the manuscript, however, the resolution of the utilized soil map is 10 km and local soil forms are not taken into account. For standardisation, the soil profile depth is fixed to 200 cm and layer discretization is fixed to 10 cm. The interactive output map presents rastered plant-available water at resolution of 1 km for different vegetation classes as well as for combined prevailing land use. Additionally, each pixel contains information on the soil moisture developments for the past month and soil profile data from the soil map. The output resolution and gapless, non-transparent raster-based presentation mode makes it difficult for end-users to navigate and impossible to identify the respective forests borders as well as to get specific and detailed information for smaller forest areas.

## 4.2 Usefulness of the operational soil moisture observations.

The incorporation of operational soil moisture measurements in an operational modelling framework is a big challenge and can lead to both beneficial and confusing outcomes. Here we are referring to station-based measurements only. Rapidly developing remote-sensing products based on satellite brightness temperature (i.e. SMAP (Das et al., 2019)) have a regular updates (2-3 days), but a typical depth of up to 5 cm on the clear ground and probably do not account for the difference in humus and mineral horizons physical properties, therefore are not useful for the forests.

The first problem is the station data itself. Here typical issues might include inadequate spatial representability (station distribution and sensor number, limited vegetation variability, heterogeneity of soils), data quality (raw data post-processing for systematic and random errors, changes in measurement setups) and data assess (negotiation of the data assessability with authorities, stability and time lag of the updates). A second difficulty arises by merging simulated and observed data and thereafter interpreting the results. Due to very high heterogeneity of soils in both horizontal and vertical dimensions and its influence on the soil moisture, direct one-to-one comparison of the time series is impossible, unless one parameterise and model exact same soil profile, which was excavated before soil sensors were installed. Moreover, depth of a certain sensor placement typically does not correspond well to the simulated soil profile layering, in the best case it appears in the middle of

one relatively thin horizon. Meanwhile, the actual footprint (vertical bandwidth) of the measured moisture for this sensor could be wider. Yet another important impact factor here is vegetation. Even advanced water balance models typically do not account for the exact positioning of plant and spatial heterogeneity of the foliage, branches and root distribution. In reality though, sensors with the same depth, but installed from e.g. both sided of the tree will often show different moisture values.

Mismatches (or perfect match as well) of simulations and measurements from the operative framework should be treated with caution. Especially conclusions from a non-expert sight could be misleading. Therefore, the comparison and interpretation of the differences should be associated with expert knowledge and consider the factors mentioned above. Despite, every operational system can profit from coupling its simulations with observations. It brings the option to validate the framework 'on-fly' directly and at the same time detect potential measurement errors from the soil moisture sensors. Furthermore, in long-term it helps to adjust and improve model parameterization.

## 5 Conclusion

The article highlights the need for high-resolution operational soil moisture monitoring in the forests of Central Germany. Existing soil moisture monitoring products possess various practice-oriented limitations, including spatial coverage and resolution issues. The presented operational point-based framework addresses major shortcomings of the existing systems taking into account local information, including meteorological (station) and geographical (i.e. aspect, slope, forest type and specific soil profile data) point data. This system allow forest managers to take targeted, local-scale measures for sustainable forest management.

The framework's technical details and its architecture are described in detail, including its all core elements such as the LWF-BROOK90 water balance model, meteorological data and its processing techniques, land cover and soil parameterization based on the National Federal Forest Inventory datasets. The framework was successfully validated using soil moisture observations from 51 stations, which resulted in summarized median Pearson correlation of 0.74. The performance was found highly variable depending of the vegetation cover and soil depth. The operational mode and website as the end-product through which the results are presented to end-users are outlined. We highlighted the effect of the local scale in understanding soil moisture dynamics, illustrating the impacts of a single precipitation event and local geographical conditions on soil moisture conditions. Further, the seasonal dynamics of soil water storage in different regions was demonstrated. Additionally we showed influence of different tree species on the redistribution of the main water balance components. The results part is closed with the introduction and showcase of the Soil Moisture Traffic Light web information platform, which represents a significant advance in the dissemination of information about soil moisture for three German regions to forest managers and stakeholders, providing an intuitive and informative tool for more effective monitoring and management of forest ecosystems. The limitations and advantages of the presented framework were discussed from an end-user perspective and compared to other existing systems. Finally, a critical outlook was given on an integration of operational soil moisture measurements into the framework.

## 6 Outlook

Since the pilot release in 2022, we continue development of the Soil Moisture Traffic Light and the framework frequently receives new upgrades. In the future, these updates could include the following features:

- Model parameterisation itself could be improved, especially the representation of LAI as one of the most sensitive model parameters. Incorporation of dynamic annual values using Coupmodel (Jansson and Karlberg, 2004) method (in-build in the LWF-BROOK90 R-package) based on MODIS satellite data will improve the evaporation
estimations.

- Operational mode of the framework could be expanded to a seasonal forecast time-scale, by getting access to German Meteorological Service long-term meteorological forecast data.

- Other tree species could be included, based on the availability of appropriate parametrization sets. The framework in general could be set up for other vegetation types. This could be especially of high interest in the case of simulations
for various crops and thus advancing agricultural management.

- Extension of the framework to the whole of Germany is possible, as the NFI database and climate data are available. For that, the computational costs will be much higher and need to be advanced.

- As a 1D-model, LWF-BROOK90 is not suited to simulate stagnant and groundwater-influenced sites in a satisfying manner. Therefore, other model (i.e. SWAP-model (Kroes et al., 2017)) or model coupling is needed in these cases.

**Data and Code availability**

Soil Moisture Traffic Light web-platform is assessable via https://life.hydro.tu-dresden.de/BoFeAm/dist_bfa_kk/index.html. Raw data used in the platform (with daily updates) could be assessed via https://life.hydro.tu-dresden.de/bfa_out/. Historical simulations (01.01.1990-14.04.2024) containing water balance components are available in HydroShare repository (Vorobevskii, 2024). Soil moisture measurement data and raw validation results could be provided upon request.

**Author contribution**

Conceptualization VI, LTT and KR; data curation VI, LTT and PR, formal analysis VI, methodology VI, LTT; visualization VI; writing: original draft preparation VI and LTT, writing: review KR, PR.

**Competing interests**

The authors declare that they have no conflict of interest.

**Acknowledgements**

Authors would like to express great thanks to Thomas Grünwald (TU Dresden), Alexander Peters (SachsenForst) and Merten Sven (ThüringenForst) for provided soil moisture measurement data.

**Funding**

Open Access Funding by the Publication Fund of the TU Dresden. This research was also funded by the German Federal
Ministry of Education and Research (FKZ 01LR 2005A—funding measure "Regional Information on Climate Action" (RegIKlim), section (a) Model Regions.

**Appendix**

### A1. Summary table on the soil moisture measurement stations

| # | Name | Source | Begin | End | Sensors and their depths | Elevation [m a.s.l.] | Precip. [mm/year] | Domin. vegetation | Soil type |
|---|------|--------|-------|-----|--------------------------|----------------------|-------------------|-------------------|-----------|
| 1 | Dillstaedt | ThueringenForst | 07.05.2015 | 09.04.2024 | 20(2), 50(2), 100(2) | 428 | 998 | spruce | Braunerde |
| 2 | Eisenberg | | 15.01.2010* | 11.04.2024 | 20(3), 35(3), 50(3), 100(2) | 764 | 1240 | spruce | Braunerde |
| 3 | Hainich | | 21.01.2020 | 13.04.2024 | 20(2), 50, 100(2) | 423 | 734 | beech | Braunerde-Terrafusca |
| 4 | Harz | | 05.05.2015 | 18.04.2024 | 20(2), 50(2), 100(2) | 565 | 985 | beech | Braunerde |
| 5 | Hohe Sonne | | 12.12.2019 | 12.04.2024 | 20(2), 50(2) | 379 | 837 | beech | Braunerde |
| 6 | Holzland | | 01.01.2000* | 19.04.2024 | 20(4), 50(4), 100(4) | 347 | 668 | pine | Braunerde-Podsol |
| 7 | Kyffhaeuser | | 13.12.2019 | 17.04.2024 | 20(2), 50(2), 100(2) | 312 | 633 | beech | Rendzina |
| 8 | Lehesten | | 15.05.2008* | 15.04.2024 | 20(3), 50 | 535 | 868 | fir | Braunerde |
| 9 | Leinawald | | 24.01.2020 | 21.04.2024 | 20(2), 50(2), 100(2) | 224 | 731 | oak | Pseudogley |
| 10 | Neuaergerniss | | 23.01.2020 | 22.04.2024 | 20(3), 50(2), 100(2) | 408 | 717 | pine | Gley-Pseudogley |
| 11 | Paulinzella | | 06.05.2015 | 14.04.2024 | 20(2), 50(2), 100(2) | 408 | 890 | pine | Braunerde-Podsol |
| 12 | Possen | | 01.01.1996* | 16.04.2024 | 20(2), 35, 50(2), 75, 100(2) | 365 | 658 | beech | Parabraunerde |
| 13 | Roemhild | | 22.04.2020 | 23.04.2024 | 20(2), 50(2), 100(2) | 315 | 799 | oak | Pelosol-Braunerde |
| 14 | Steiger | | 01.01.2001* | 09.04.2024 | 20(2), 50(2), 100(2) | 299 | 625 | oak | Parabraunerde |
| 15 | Vessertal | | 11.12.2019 | 09.04.2024 | 20(2), 50(2), 100(2) | 775 | 1240 | beech | Braunerde |
| 16 | Grillenburg | FLUXNET/ICOS | 09.11.2006* | 31.12.2023 | 10 (3), 20, 30, 40, 50 | 385 | 873 | grass | Gleysol |
| 17 | Hainich | | 01.03.2000 | 20.11.2023 | 8, 16, 32 | 438 | 714 | beech | Stagnic Cambisol |
| 18 | Hetzdorf | | 17.12.2017* | 31.12.2023 | 10 (2), 20, 30, 40, 50 | 395 | 871 | oak | Stagnic Albeluvisol |
| 19 | Hohes_Holz | | 01.01.2015* | 31.12.2023 | 5 (2), 15 (2), 25 (2), 35 (2), 50 (2), 70 (2) | 193 | 635 | oak | Luvisol (Fahlerde) |
| 20 | Leinefelde | | 18.04.2002 | 31.12.2012 | 8, 16, 32, 62 | 453 | 743 | beech | Leptosol/Luvisol |
| 21 | Obernbaerenburg | | 06.05.2009 | 31.12.2014* | 10, 20, 35 | 734 | 1052 | spruce | Normpodsol |
| 22 | Tharandt | | 24.02.1997* | 31.12.2023 | 5, 10 (2), 20, 50 | 385 | 829 | spruce | Cambic podzol |
| 23 | Altenberg | SachsenForst | 01.01.2022 | 04.04.2024 | 20 (3), 60(2), 100 | 750 | 1052 | spruce | Podsol-Braunerde |
| 24 | Bautzen | | 01.01.2022 | 31.03.2024 | 20 (3), 60 (2), 80 | 440 | 719 | spruce | Braunerde-Podsol |
| 25 | Colditz | | 11.01.2007* | 02.04.2024 | 10 (5), 20(3), 30 (5), 60 (6), 100 | 200 | 717 | oak | Pseudogley |
| 26 | Cunnersdorf | | 14.10.2021 | 31.03.2024 | 20 (3), 60 (2), 100 | 440 | 861 | spruce | Braunerde-Podsol |
| 27 | Klingenthal | | 01.10.2021 | 31.03.2024 | 20 (2), 60 (2), 100, 200 | 840 | 1094 | spruce | Podsol Braunerde |

| 28 | Laussnitz | | 28.04.1994* | 01.04.2024 | 10 (4), 20 (3), 30 (9), 60 (6), 100 | 170 | 744 | pine | Braunerde-Podsol |
|----|-----------|--|-------------|------------|-------------------------------------|-----|-----|------|------------------|
| 29 | Nationalpark | | 11.11.2010* | 03.04.2024 | 10 (3), 20(3), 30 (2), 60 (3), 100 | 260 | 847 | beech | Braunerde |
| 30 | Olbernhau | | 06.10.2021 | 31.03.2024 | 20 (3), 60 (2), 100 | 710 | 969 | spruce | Braunerde-Podsol |
| 31 | Bautzen | | 09.01.2018 | 02.05.2024 | 30 | 359 | 719 | grass | Braunerde-Podsol |
| 32 | Colditz | | 10.01.2007 | 04.05.2024 | 30 | 183 | 717 | grass | Pseudogley |
| 33 | Cunnersdorf | | 12.01.2017 | 01.05.2024 | 30 | 459 | 862 | grass | Braunerde-Podsol |
| 34 | Doberschuetz | | 21.09.2004 | 29.04.2024 | 30 | 104 | 661 | grass | reliktischer Gley-Podsol |
| 35 | Eich | | 12.01.2017 | 10.05.2024 | 30 | 469 | 946 | grass | Parabraunerde-Pseudogley |
| 36 | Graupa | | 12.01.2017 | 06.05.2024 | 30 | 196 | 833 | grass | Normhortisol |
| 37 | Heinzebank | | 12.01.2017 | 08.05.2024 | 30 | 615 | 960 | grass | Normbraunerde |
| 38 | Klingenthal | | 12.01.2017 | 29.04.2024 | 30 | 863 | 1094 | grass | Braunerde-Podsol |
| 39 | Laussnitz | | 12.12.2006 | 03.05.2024 | 30 | 184 | 744 | grass | Braunerde-Podsol |
| 40 | Malschwitz | | 03.05.2016 | 03.05.2024 | 30 | 147 | 719 | grass | Brauneisengley |
| 41 | Nationalpark | | 12.01.2017 | 05.05.2024 | 30 | 325 | 847 | grass | Normparabraunerde |
| 42 | Neukollm | | 01.01.2018 | 01.05.2024 | 30 | 153 | 720 | grass | Eisenpodsol |
| 43 | Nochten | | 06.10.2015 | 02.05.2024 | 30 | 134 | 701 | grass | podsoliger Gley |
| 44 | Olbernhau | | 12.01.2017 | 30.04.2024 | 30 | 704 | 971 | grass | Braunerde-Podsol |
| 45 | Rittersgruen | | 12.01.2017 | 09.05.2024 | 30 | 770 | 991 | grass | Normbraunerde |
| 46 | Roitzsch | | 06.12.2006 | 29.04.2024 | 30 | 127 | 667 | grass | Parabraunerde-Pseudogley |
| 47 | Schlottwitz | | 12.01.2017 | 07.05.2024 | 30 | 369 | 865 | grass | Normbraunerde |
| 48 | Schoenheide | | 12.01.2017 | 11.05.2024 | 30 | 634 | 1038 | grass | Braunerde-Podsol |
| 49 | Werdau | | 01.01.2012 | 05.05.2024 | 30 | 381 | 817 | grass | Parabraunerde-Pseudogley |
| 50 | Wermsdorf | | 31.05.2013 | 30.04.2024 | 30 | 216 | 639 | grass | Pseudogley-Fahlerde |
| 51 | Zellwald | | 19.09.2008 | 04.05.2024 | 30 | 338 | 814 | grass | Auengley |

\* begin and end periods of the measurements significantly vary for different moisture sensors

**A2. Potential availability of the meteorological data from German Meteorological Service for the last ten years**

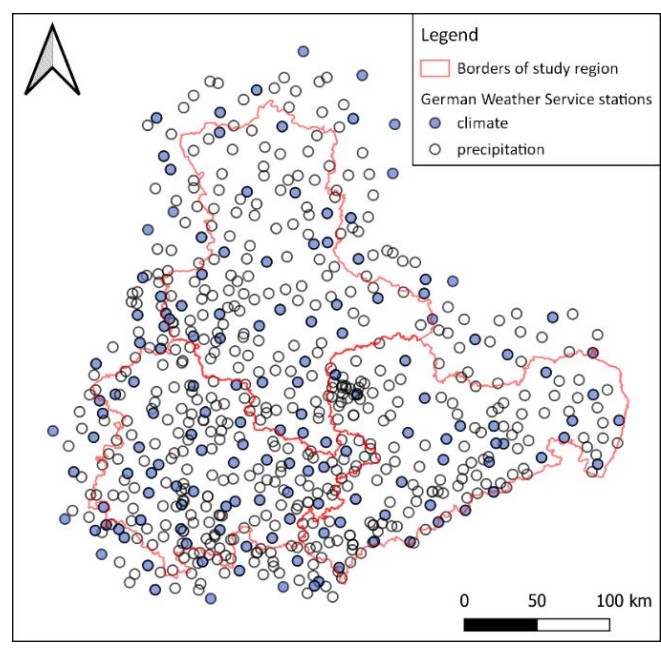

## A3. LWF-BROOK90 Vegetation-specific parameters

| Parameter | Description | Unit | Beech | Oak | Spruce | Pine | Grass |
|---|---|---|---|---|---|---|---|
| lwidth | Average leaf width | m | 0.04 | 0.05 | 0.001 | 0.0015 | 0.005 |
| rhotp | Ratio of total leaf area to projected area | - | 2 | 2 | 2.6 | 2.6 | 2 |
| cintrl | Maximum interception storage of rain per unit LAI | - | 0.6 | 0.7 | 0.4 | 1 | 0.1 |
| cintrs | Maximum interception storage of rain per unit SAI | - | 0.5 | 1 | 0.2 | 1 | 0 |
| cintsl | Maximum interception storage of snow per unit LAI | - | 2.4 | 2.8 | 1.6 | 4 | 0.1 |
| cintss | Maximum interception storage of snow per unit SAI | - | 2 | 4 | 0.8 | 4 | 0 |
| frintlai | Intercepted fraction of rain per unit LAI | - | 0.08 | 0.1 | 0.08 | 0.13 | 0.08 |
| frintsai | Intercepted fraction of rain per unit SAI | - | 0.08 | 0.1 | 0.08 | 0.13 | 0 |
| fsintlai | Intercepted fraction of snow per unit LAI | - | 0.08 | 0.1 | 0.08 | 0.13 | 0.08 |
| fsintsai | Intercepted fraction of snow per unit SAI | - | 0.4 | 0.5 | 0.1 | 0.3 | 0 |
| alb | Albedo of soil/vegetation surface without snow | - | 0.21 | 0.15 | 0.1 | 0.15 | 0.25 |
| albsn | Albedo of soil/vegetation surface with snow | - | 0.47 | 0.5 | 0.55 | 0.4 | 0.4 |
| obsheight | Mean height of obstacles on soil surface | m | 0.02 | 0.02 | 0.02 | 0.02 | 0.02 |
| height | Plant height | m | 30 | 25 | 30 | 30 | 0.5 |
| maxlai | Maximum projected leaf area index | m2 m-2 | 6 | 4.5 | 6.5 | 3.5 | 3.5 |
| sai | Steam area index | m2 m-2 | 1 | 0.9 | 2 | 0.8 | 0 |
| winlaifrac | Minimum LAI as a fraction of maxlai | - | 0.1 | 0.1 | 0.9 | 0.8 | 0.1 |
| cs | Ratio of projected stem area index to canopy height | m-1 | 0.035 | 0.035 | 0.035 | 0.035 | 0.035 |
| radex | Extinction coefficient for solar radiation and net radiation in the canopy | - | 0.59 | 0.59 | 0.45 | 0.45 | 0.7 |
| glmax | Maximum leaf vapour conductance when stomata are fully open | m s-1 | 0.006 | 0.007 | 0.0035 | 0.0045 | 0.008 |
| glmin | Minimum leaf vapour conductance when stomata are closed | m s-1 | 0.0002 | 0.0003 | 0.0001 | 0.0002 | 0.0001 |
| maxrootdepth | Maximum root depth (positive downward) | m | -1.6 | -2 | -1.2 | -2 | -0.5 |
| betaroot | Shape parameter for rootlength density depth distribution | - | 0.966 | 0.966 | 0.976 | 0.976 | 0.943 |
| age_ini | Age of stand (for root development) | a | 50 | 100 | 100 | 100 | 1 |
| maxrlen | Total length of fine roots per unit ground area | m m-2 | 3200 | 3200 | 2000 | 2500 | 600 |
| psicr | Critical leaf water potential at which stomata close | MPa | -2 | -2.5 | -2 | -2.5 | -1.5 |
| rrad | Average radius of the fine or water-absorbing roots | mm | 0.25 | 0.25 | 0.25 | 0.25 | 0.25 |

## A4. Additional validation results

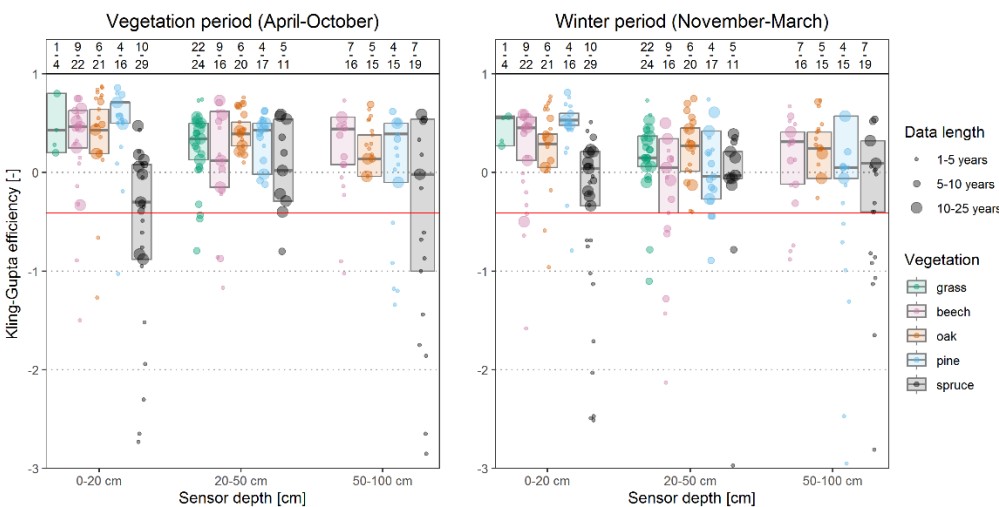

**Figure A4.1. Daily Kling-Gupta-Efficiency between simulated and observed soil moisture for all stations and sensors. Boxplots without whiskers represent 25-75% quantile range and median calculated from weighted values (after data length). Red line with KGE= –0.41 represents critical value below which the modelled time-series are not better than mean value. Numbers above each category state for the number of stations (numerator) and sensors (denominator).**

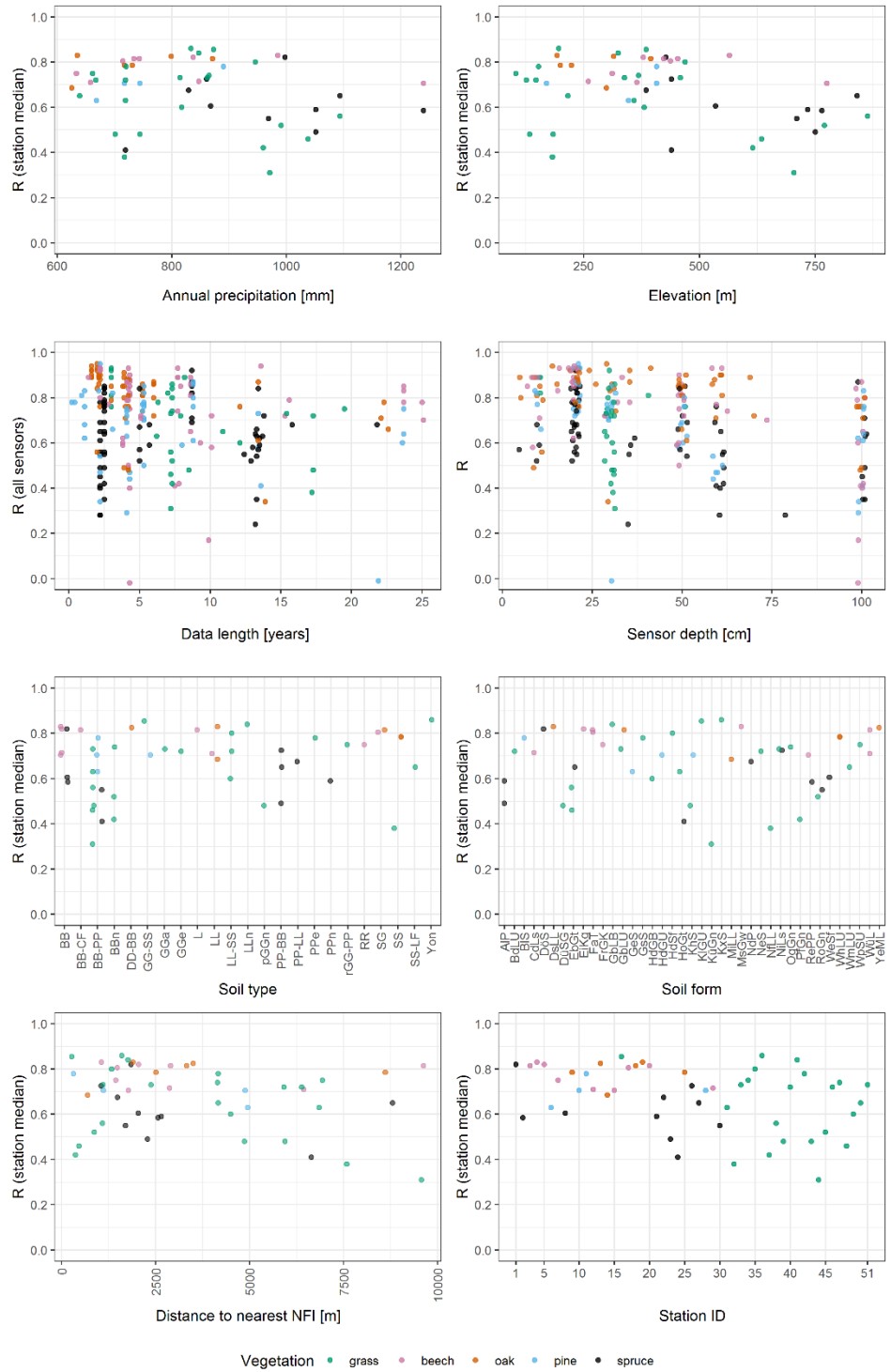

Figure A4.2. Pearson correlation between simulated and observed soil moisture for all stations and sensors (daily scale) against site characteristics.

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
