# Peer review of "High-resolution operational soil moisture monitoring for forests in the Middle Germany"

_Hydrology and Earth System Sciences, 2023_

## Referee Comment (RC2)

OVERVIEW:

The paper aims to present in detail a soil moisture monitoring for the forests of the Middle Germany focusing on presenting the website based platform hosting the monitoring framework. The methodological framework is mostly well described and justified throughout the manuscript. The available data and the capabilities to display soil moisture status over a large forest territory is without a doubt valuable for stakeholders, forest managers and scientists. Moreover, the paper is generally well written (I especially liked the introduction). From reading the manuscript it is easy to see that the author team is very knowledgeable about their system and have taken the time to describe it and justify most of the choices within the framework architecture.

However, at its current state the manuscript appears to me not completely useful for non of the collectives you aim to address (i.e. international scientific community or stakeholders and forest managers at the national level). Below I outline why, some suggestions as well as specific comments on text and figures.

GENERAL COMMENTS

What would make the manuscript more interesting/engaging to the international scientific community?

(I) **Improve readability**: there are multiple acronyms throughout the text (and also in the figures) which are not always easy to remember (reader needs to search through the document). Some of the explanations are very specific for forest managers or forest researcher (e.g. see my comment on track corners). I think the soil moisture monitoring system you have developed is valuable and very interesting.

(II) **Scientific discussion and placing your SM monitoring within the context of available literature and ongoing SM monitoring efforts elsewhere**: I miss more discussion of how the system you present is "an operational high-resolution soil moisture monitoring framework for the forests in Middle Germany, which addresses the main limitations and problems of the existing monitoring systems". What are the existing monitoring systems and what are their shortcomings? Could you include a discussion on that? Also, are you referring to existing monitoring systems in Germany or around the world? What would be a comparably good SM monitoring system in another country? An example of an SM monitoring systems, based on SM observations from cosmic ray neutron sensing would be COSMOS UK (see https://cosmos.ceh.ac.uk/data ). They feature a similar system to the traffic light system you describe, you could perhaps compare it to such a system and/or include examples from forested sites.

**Discussion on soil moisture observations and their usefulness in such a SM monitoring framework**: There is no mention or discussion on how to incorporate actual soil moisture observations in your framework and how it would benefit from it. Are there any hydrological observatories where you could apply your modelling framework but then improve it? You make a shy suggestion in the Outlook section, but that is rather short and underdeveloped.

**Perhaps too much German on figures and in text**: in Section 4.4. as a non-proficient German speaker I found it difficult and unmotivating to follow. I would find it much more interesting, if the platform could be presented in the publication already with translation in English (i.e. wait until then to publish this

contribution or state a date, ideally in the near future, when the website will be available in English). Alternatively you can take the focus away from the online platform and mention it briefly and also produce a short video tutorial in English for users interested in the data and science behind it. Then focus much more on discussing the science (see general comment ii).

What would make the manuscript more interesting/engaging to stakeholders, environmental authorities and forest managers at the national level?

(i)     If the platform is intended to engage more stakeholders and forest managers in Germany, I believe it would be much more beneficial to publish in a German scientific journal which is also easily available for environmental authorities and forest managers. This is also where German speaking scientists interested in the platform (again because it is only in German at the moment) can explore it. At the moment the only way to go through the different options is via translating the page. When you do that the images stall and the page takes longer to load. For the expert mode you need to know at least some German or be patient to translate to start using the data files downloaded.

(ii)    To address such a public, perhaps also the text would need to be rewritten and more emphasis on how to use the platform and perhaps a couple of examples of the benefits of using it (i.e. practical examples) should be included.

To summarise, I think the SM monitoring framework presented here is very interesting and valuable as well as it has constituted a great effort to produce and should be shared within the international scientific community. However, I think that currently the paper is not suitable for HESS and should find its place in a different journal. For that I suggest either major revisions with a possible change of scope or a submission to a different journal.

Besides, see my specific comments, line by line, below:

Figure 1: Add a small inlet of Germany in one of the corners. Complement the Figure 1 caption with the meaning of the 3206 BWI abbreviation to aid readers. Briefly explain what the black dots mean (I understand is the inventory but please make it explicit).

Line 128: Have a very brief explanation of what REST API access is (few words)

Line 139: Section 3.1. is well documented/ choices well explained. However, I suggest a more intuitive sub-header starting with the model type and then introducing the name. spell out that it is a soil hydrological model and it is 1D.  Line 140: I would start the paragraph with saying what the model is about and then go into these details for the benefit of readers who are not familiar.

Figure 2: explain in the legend or caption what KL or RR stand for. Make the dot for the BWI sites slightly larger on the legend. Why is this figure relevant to show here and why not in Annexes?

Figure 3: same comment on the BWI dot

Figure 4: remove "violin plots with" from figure caption, it is redundant. Otherwise figure is quite informative

Figure 5: useful figure giving a good overview. Small detail on caption, change to "for a selected"

Figure 6: green balloon "Daily meteostation data from 2010" sounds like the data is from that year. I understand it is from 2010 onwards and up to current?

Section 4.1. Line 272 what does "first hundreds of meters" mean in this sentence? please rephrase or clarify. Also you could discuss the differences between the point and raster set up already in Methods (I did not see it there). This section is not so easy to read, I expect more documenting (i.e. references and comparison between raster and point set ups).

Line 274 typo "in" instead of "is"

Figure 7: On your (b) plot in the legend the light yellow and green are very difficult to see. You have the same issue on the lightest colours in Figure 8 on the legend.

Fig 9: very nice and informative on the evolution of SM along a whole year

Line 357: for the readers who may not know what track corners are, can you please include a reference?

Figure 10: another interesting and useful figure from scientific point of view.

Figure 11 and 12: entirely in German, basically snapshots and (at least when I download the pdf), the resolution is quite low. I struggle to see the text (Fig 11 for example) and think it occupies unnecessary space. Instead of these figures an explanatory video could be much more useful.

---

## Author Comment (AC2)

| # | Comment | Answer |
|---|---------|--------|
| 1 | MAJOR: The authors have developed a point-scale modelling system and have emphasised throughout the paper that this type of analysis has several advantages over a grid-based modelling system. Point-scale analysis can have some advantages, but with the current availability of computing and storage facilities, grid-based analysis has become very feasible even at 30 m resolution (e.g. Vergopolan et al., 2021, https://doi.org/10.1038/s41597-021-01050-2). Furthermore, point scale analysis does not consider lateral movement of water, which is particularly important over shallow soils and sloping terrain. Therefore, grid-based systems have also advantages with respect to point-scale simulations. The discussion of the limitations of the proposed approach should be clarified and the comparison with grid-based analysis should be made more fair. | Agreed partly. We will add a discussion section on point vs grid based soil moisture simulations for forest areas with regard to resolution, data input, pros and cons, plausibility and user orientation. A few extra features of the point-based system which are not mentioned in the text include the following ones. For forest managers the actual data availability at the point (particularly detailed soil information and its near real time as well as retrospectacle hydrological behaviour) is much more of usage as the generalised grid-based information, everything else is often redundant. Further, point modelling has the advantage that we can update point information (input parameters) and/or add new points (soil profiles) easily and at any time without having to dismantle the entire model concept. The point approach therefore offers an open space approach for continuous improvement. Point display has more practical advantages because additional information from the site (soil profile and soil properties, climate summary, topography information) fits into the display, not to forget the helpful background (aerial, OSM, forest site maps etc). On the other hand, grid-based simulations are easily for postprocessing and comparison to other similar system outputs. Regarding grid-based soil moisture estimates from Vergopolan et al. 2021. This is a reanalysis dataset (with application of modelling and several stages of downscaling techniques), which is not a daily operational framework, therefore is not comparable regarding the computational powers behind. Further, data from the satellite sensors used in the study typically have a depth penetration of up to 5 cm on the clear ground and do not work in the forested areas, not to mention, that probably the setup and chosen soil parameterisation dataset did not account for the difference in humus and mineral horizons physical properties in forests. Yes, the used 1D model could not adequately represent lateral flows (especially inflow) on the sloped terrain and we will address it in the discussion. However, this is not so critical for the |

| # | Comment | Answer |
|---|---------|--------|
| | | study sites, since approximately 75% of forest floors (at least in Saxony) are characterised by predominant vertical seepage water movement and here LWF BROOK90 works better than, for example, the grid-based SWAT or TOPMODEL. Finally, lateral inflows and outflows, which are the main advantage of grid-based models, have to use a good quality DTM. In Central Europe, DEMs (and therefore streamflow networks) are massively covered with artefacts due to the strong anthropogenic influence (bridges, forest roads, mining traces, dams, culverts, ditches) and would have to be extensively cleaned up in order to actually achieve the alleged accuracy of soil moisture modelling, which is problematic for dense vegetation covered areas. Finally, accurate correction for tree height to produce high-resolution DTM is possible only with laser/drone scans due to high heterogeneity of the forest stand. |
| 2 | MAJOR: In the abstract it reads "soil moisture monitoring framework … which addresses the main limitations and problems of the existing monitoring systems.". This is not demonstrated at all in the paper. | Agreed, we elaborate this in a new discussion section (see also comment below). As the primary stakeholders and users of the system are forest managers of different administrative levels, we gather direct feedback from them. And they agreed that the existing systems are too coarse in resolution, which was the main point of the development of this point-based modelling system with point-based soil information data from National Forest Inventory and high-resolution local soil map (1:10 000, where the smallest area-mapping unit is about 0.5 ha), rather than generalised soil maps of coarser resolution (1:1 000 000). Nevertheless, there are hardly any directly quotable sources to be found to this problem (only 'practical experience' and 'expert knowledge') and we could only summarise and mention this feedback. From the point of view of soil hydrology, however, there is also a need to map the high spatio-temporal heterogeneity of soil moisture, which is largely controlled by the local variability of physical soil properties, topography and |

| # | Comment | Answer |
|---|---------|--------|
| | | vegetation. (e.g. https://doi.org/10.1029/2022WR032719 or https://doi.org/10.1016/j.foreco.2016.12.024 or https://doi.org/10.1016/j.foreco.2020.118671). Therefore, our approach is to link the data available at the plot level with the best available model parameters and modelling techniques at the same spatial scale. |
| 3 | MAJOR: Two soil moisture modelling systems were mentioned in the introduction, the German Drought Monitor and the German Weather Service Soil Moisture Viewer. These systems have high resolution, 4 km and 1 km respectively, not so different from the point modelling system developed in this study, note that the average distance of the rain gauge is 5 km and more for the other meteorological data. I believe that a comparison should be made between the simulation carried out in this study and these systems. Through this analysis, the potential added value of the developed monitoring system can be evaluated. | Agreed partly. The more detailed comparison will be added in the new discussion section, while information in the introduction will be reduced (current L55-65). All three systems use more or less the same meteorological data (main difference are quality-control and interpolation techniques). Currently, the online version of German Drought Monitor uses the MHm model, which possesses limitations with regard to evaporation representation and 1:1 000 000 resolution soil map. The German Weather Service Soil Moisture Viewer uses LWF-BROOK90 (same as in the presented setup) model for forests and a 1:1 000 000 resolution soil map. The system was updated after the manuscript submission (before it was only for croplands).
It is questionable whether the average distance of 5 km to the rain gauge dictates the resolution of the product. For instance, German Drought Monitor uses regular-grid-interpolation inputs and has no variability due to soil or meteorological datasets within a grid (4 km), while the presented framework will still possess point-wise variability due to higher resolution of soil dataset and variable distance to the meteostations for each point. Exact build-up of the second system is unfortunately unknown, but presumably the above mentioned statement holds for it as well (at least regarding variability due to soil dataset). Raster-based simulation at the resolution of the used soil dataset will require a grid of 10 m, which is implausible for the implementation due to computational power requirement and technical problems of the result representation (in an interactive and not just raster-based way). |

| # | Comment | Answer |
|---|---------|--------|
|   |         | The raw simulation data from the two systems is not open-sourced (only post-processed and converted in indexes and presented via picture-based format). We will make requests to corresponding authors of these two systems. If the data will not be available, we will be able to make only a qualitative analysis and discussion of the setups, which from our point of view will still be enough to conclude on the potential added value of the presented framework. |
| 4 | MAJOR: A major and important issue is the lack of validation. Only one figure, not discussed, with a comparison to a single soil moisture station. In addition, the paper also reports long-term averages of evaporation per forest type. How accurate are the evaporation estimates? How accurate are the soil moisture estimates? In order to be published, the paper should perform a robust validation of the simulations. | Agreed partly, the main purpose of the presented paper is not about robust validation, but about presenting the system itself. Besides, it is mentioned in the text (L 236-239), that the soil moisture pilot version of the setup was already validated (for different forest types and grass) based on soil moisture measurements in Saxony (http://dx.doi.org/10.1127/metz/2023/1155). Additionally, we want to point out, that due to high heterogeneity of the soils in the region, direct comparison of measured soil moisture with simulations from nearby BWI point could lead to unexpected results, even if the vegetation above is completely the same, due to possible differences in soil structure and profile depth. Evaporation was validated similarly to the presented setups (with original BROOK90 model, which has the same evaporation module) for 5 eddy-covariance towers in Saxony (https://doi.org/10.5194/hess-26-3177-2022). Nevertheless, we will add an extra section to the results with validation of the soil moisture and evaporation components using available data of approximately 15 stations (forest climate stations and FLUXNET/ICOS towers) and nearby simulated BWI points. |

| # | Comment | Answer |
|---|---------|--------|
| 5 | MODERATE: The text contains several errors, and some parts are unclear (see specific comments for some of them). Acronyms are not defined. Many references are in German. Four figures are in German. This can't be accepted in an international journal. The figures should be translated into English and the German references should be avoided or minimised. | Agreed, unclear parts and errors mentioned below will be revised and specific acronyms will be checked and their number reduced to minimum. Figures 13 and 14 (current numbering) will be erased since new sections are planned (i.e. validation) and the platform will be illustrated with updated Figures 11 and 12 (current numbering). Either the Figures themselves will be translated or the platform becomes an English lite version, we are considering technical solutions for this. |
| 6 | MODERATE: The model is described in Section 3.1. Several modules are mentioned, e.g. vegperiod, betamode, b90, ... However, a detailed description of these modules is missing. The reader is lost and, for example, it's unclear how the values of the model parameters were assigned. How many parameters? The model description needs to be improved. | Agreed partly, the main principles of mentioned modules are presented in the text (L150-152 and L161-166). We will revise and enhance to a certain extent the description, however providing a full description of them in the text is unnecessary from our point of view, as the potential reader could always refer to a provided reference and it shifts the focus of the section. Additionally, we suggest putting a 'summary' table on the model parameterization in the Appendix. |
| 7 | L71: "with an operational climate data". It's not climate, but meteorological data. To be corrected throughout the paper. | Agreed, will be corrected and checked throughout the text. |
| 8 | Figure 1 caption: Specify acronyms. | Agreed, will be added. |
| 9 | L92: The size of the investigated area should be specified. | Agreed, will be added. It is about 55000 km2. |
| 10 | L128: What is the "REST-API access"? | Agreed, will be clarified. |
| 11 | L135-136: The sentence is unclear, and it should be revised. | Agreed, will be revised. |
| 12 | L140: What does LWF stand for? Check all acronyms. | Agreed, will be added. LWF stands for 'Landesanstalt für Wald und Forstwirtschaft' Bayern, where the original model was first modified. |
| 13 | L188: Why are some stations filtered? What do the authors mean with "filtered"? | Agreed, will be clarified. Station list available for the whole of Germany was reduced based on spatial (study region plus 30 km buffer zone) and |

| # | Comment | Answer |
|---|---------|--------|
| 14 | L190: Which criteria should be matched? | temporal (data availability within the last 10 years period) principles. |
| 15 | L191-193: The sentence is unclear, and it should be revised. | Agreed, will be clarified. |
| 16 | L201: It's not clear for which period the soil moisture data are simulated from the system. Here it reads 5 months, later in the text 30-year period. It must be clarified. | Agreed, will be clarified. In operational mode we simulate 10 years each day. 30-years simulations were done once to calculate quantile REW values for each plot. Here we mean that tests of the meteorological data retrieval for 10-year simulations during the June-October 2023 period did not show large deviations in station data availability. This means that up-to-date data for each used variable is almost stable regarding the number of stations. |
| 17 | L08-209: Approximate distance for rain gauge equal to 5 km, more than 10 km for other meteo data. The actual resolution of the simulated soil moisture cannot be less than 5 km. | Agreed partly, see comment #3. The resolution of the meteorological forcing does not necessarily dictate the resolution of the soil moisture simulations. For each simulated point within the same BWI plot, meteorological input will be different due to different distances, even if the original station data picked by the filter is the same. Moreover, each point has its own geographical features, including topography and soils. These two factors bring enough variability and confirm the higher output resolution compared to grid-based products. Furthermore, based on our previous study for the evaporation component, it was found that the variability and uncertainty of the parameterization datasets is higher than the uncertainty of the meteorological data (https://doi.org/10.5194/hess-26-3177-2022). |
| 18 | L226-227: It's unclear if NA values are present or not in the data. Please revise the sentence. | Agreed, will be clarified. So far we did not experience such a case, while for each plot
1) stations in 30 km buffer with more than 5% on NA are removed
2) typically more than one station for plot inside the buffer
3) for 1-day-lag data 30 km buffer is expandable if NA appear
So typically, after all these steps there is no NA data inside. However, if it happened (i.e. for wind |

| # | Comment | Answer |
|---|---------|--------|
| | | or sunshine duration data, as the station network is sparse; or there is an update failure by certain DWD stations) this simplified gap-filling algorithm was added out of precaution, so 100% no NA meteorological data appear in the forcing. |
| 19 | L233-235: The sentence is unclear, and it should be revised. | Agreed, will be revised. |
| 20 | L270: I don't agree that raster-based simulations do not account for local conditions, it depends on the grid size. Please revise (see the first general comment). | Agreed, will be revised. |
| 21 | L273-274: I don't believe the two examples "illustrate the advantage of the point-based framework". This part should be revised. | Agreed, will be revised. |
| 22 | L335: What is the time period of the long-term simulations? | Agreed, will be added. Time-period is 1990-2020. |

---

## Author Comment (AC3)

| # | Comment | Answer |
|---|---------|--------|
| 1 | Improve readability: there are multiple acronyms throughout the text (and also in the figures) which are not always easy to remember (reader needs to search through the document). Some of the explanations are very specific for forest managers or forest researcher (e.g. see my comment on track corners). I think the soil moisture monitoring system you have developed is valuable and very interesting. | Agreed, specific acronyms will be reduced to minimum. |
| 2 | Scientific discussion and placing your SM monitoring within the context of available literature and ongoing SM monitoring efforts elsewhere: I miss more discussion of how the system you present is "an operational high-resolution soil moisture monitoring framework for the forests in Middle Germany, which addresses the main limitations and problems of the existing monitoring systems". What are the existing monitoring systems and what are their shortcomings? Could you include a discussion on that? Also, are you referring to existing monitoring systems in Germany or around the world? What would be a comparably good SM monitoring system in another country? An example of an SM monitoring systems, based on SM observations from cosmic ray neutron sensing would be COSMOS UK (see https://cosmos.ceh.ac.uk/data ). They feature a similar system to the traffic light system you describe, you could perhaps compare it to such a system and/or include examples from forested sites. | Agreed. Existing systems we found are briefly mentioned in the introduction. The more detailed comparison will be added in the new discussion section, while information in the introduction will be reduced (current L55-65), as also requested by Reviewer #1. We are referring mainly to European systems, but we will continue our search to find, mention and compare similar systems worldwide. Regarding the COSMOS UK system you've mentioned. It is a very nice example of a monitoring system using observations. We will definitely mention it in comparison. However, as the observations are very sparse, they are unable to cover larger areas as well as to be representative with regard to spatial soil heterogeneity. In addition, coupled with absence of additional information about the site itself, it is hardly applicable for the forest management. |
| 3 | Discussion on soil moisture observations and their usefulness in such a SM monitoring framework: There is no mention or discussion on how to incorporate actual soil moisture observations in your framework and how it would benefit from it. Are there any hydrological observatories where you could apply your modelling framework but then | Agreed, will be elaborated. At the present, operational soil moisture observations are available only for Saxony (forest climate stations, measurements under grass vegetation), which are already pre-processed and delivered through a third-party data provider (Pikobytes GmbH). Unfortunately, we do not have access to similar operational |

| # | Comment | Answer |
|---|---------|--------|
| | improve it? You make a shy suggestion in the Outlook section, but that is rather short and underdeveloped. | soil moisture data (forest climate stations) in Thuringia and Saxony-Anhalt to integrate it in the framework. We are still trying to negotiate it. In general, every similar system will definitely profit from coupling its simulations with observations, where it is possible. This helps not only to validate it 'on-fly', but in long-term to improve parameterization. |
| 4 | Perhaps too much German on figures and in text: in Section 4.4. as a non-proficient German speaker I found it difficult and unmotivating to follow. I would find it much more interesting, if the platform could be presented in the publication already with translation in English (i.e. wait until then to publish this contribution or state a date, ideally in the near future, when the website will be available in English). Alternatively you can take the focus away from the online platform and mention it briefly and also produce a short video tutorial in English for users interested in the data and science behind it. Then focus much more on discussing the science (see general comment ii). | Agreed, we plan to shift focus a bit from the web-platform description more in the framework description. Further, we plan to add a validation subsection in results and discussion section on the comparison of our platform to other existing systems. Nevertheless, at the same time we are exploring the technical solutions of adding an English version to the website. Alternatively, several browsers have a page translation plugin, which works well and translates the graphical part as well. We will shortly correct our German version text of the webpage and every caption labelling, so that the automatic translation will be almost correct. Thereafter we will test how long it will take for different browsers and plugins. Additionally, we plan to record tutorials for potential english-speaking users. Despite the fact, that the main users of the system are german-speaking forest managers, we believe the platform itself and the produced data could be of interest to the international community: both national forest authorities (i.e. to learn and build similar systems) and scientific community (i.e. in research of forest ecosystems dynamics and variability). The pilot version of the system was already presented for German-speaking users in parallel via various application-oriented journals and internet platforms. |
| 5 | If the platform is intended to engage more stakeholders and forest managers in Germany, I believe it would be much more beneficial to publish in a German scientific journal which is also easily available for environmental authorities and forest managers. This is also where German speaking scientists interested in the platform (again because it is only in German at the moment) can explore it. At the moment the only way to go through the different options is via translating the page. When you do that the images stall and the page takes longer to load. For the expert mode you need to know at least some German or be patient to translate to start using the data files downloaded. | |

| # | Comment | Answer |
|---|---------|--------|
| 6 | To address such a public, perhaps also the text would need to be rewritten and more emphasis on how to use the platform and perhaps a couple of examples of the benefits of using it (i.e. practical examples) should be included. | Agreed, we will elaborate on the current section 4.4 in order to address the practical part, with examples from forest managers. |
| 7 | Figure 1: Add a small inlet of Germany in one of the corners. Complement the Figure 1 caption with the meaning of the 3206 BWI abbreviation to aid readers. Briefly explain what the black dots mean (I understand is the inventory but please make it explicit). | Agreed, will be corrected and added. |
| 8 | Line 128: Have a very brief explanation of what REST API access is (few words) | Agreed, will be added. |
| 9 | Line 139: Section 3.1. is well documented/ choices well explained. However, I suggest a more intuitive sub-header starting with the model type and then introducing the name. spell out that it is a soil hydrological model and it is 1D.  Line 140: I would start the paragraph with saying what the model is about and then go into these details for the benefit of readers who are not familiar. | Agreed, will be corrected. |
| 10 | Figure 2: explain in the legend or caption what KL or RR stand for. Make the dot for the BWI sites slightly larger on the legend. Why is this figure relevant to show here and why not in Annexes? | Agreed, will be added. As we will add more (sub)sections, text and figures to the article, we move this figure in Appendix. |
| 11 | Figure 3: same comment on the BWI dot | Agreed, will be corrected. |
| 12 | Figure 4: remove "violin plots with" from figure caption, it is redundant. Otherwise figure is quite informative | Agreed, will be corrected. |
| 13 | Figure 5: useful figure giving a good overview. Small detail on caption, change to "for a selected" | Agreed, will be corrected. |

| # | Comment | Answer |
|---|---------|--------|
| 14 | Figure 6: green balloon "Daily meteostation data from 2010" sounds like the data is from that year. I understand it is from 2010 onwards and up to current? | Agreed, it was done for space saving on the chart. We suggest changing it simply to 'Meteorological data' with explanation in text, that data is from 2010 onwards. |
| 15 | Section 4.1. Line 272 what does "first hundreds of meters" mean in this sentence? please rephrase or clarify. Also you could discuss the differences between the point and raster set up already in Methods (I did not see it there). This section is not so easy to read, I expect more documenting (i.e. references and comparison between raster and point set ups). | Agreed, this will be part of a new discussion section, as it was also pointed out by Reviewer #1. |
| 16 | Line 273 typo "in" instead of "is" | Agreed, will be corrected. |
| 17 | Figure 7: On your (b) plot in the legend the light yellow and green are very difficult to see. You have the same issue on the lightest colours in Figure 8 on the legend. | Agreed, we will try other common colour-blind schemes to improve visibility. |
| 18 | Fig 9: very nice and informative on the evolution of SM along a whole year | Thank you for pointing this out. |
| 19 | Line 357: for the readers who may not know what track corners are, can you please include a reference? | Agreed, will be clarified in Section 2.2 |
| 20 | Figure 10: another interesting and useful figure from scientific point of view. | Thank you for pointing this out. |
| 21 | Figure 11 and 12: entirely in German, basically snapshots and (at least when I download the pdf), the resolution is quite low. I struggle to see the text (Fig 11 for example) and think it occupies unnecessary space. Instead of these figures an explanatory video could be much more useful. | Agreed, Figures 13 and 14 (current numbering) will be erased since new sections are planned (i.e. validation) and the platform will be illustrated with updated Figures 11 and 12 (current numbering). Either the Figures themselves will be translated or we present or the platform becomes an English lite version, we are thinking about technical solutions for this. A tutorial video is a very nice idea, we will definitely consider embedding it on the website. |

---

## Author Response (AR1)

**1. Editor comments**

| # | Comment | Answer | Changes |
|---|---------|--------|---------|
| 1 | The REW variable is a key indicator in your study. However, the definitions and use of the soil-water content at the conditions of "field capacity" and "permanent wilting" are presented in a rather inadequate manner. First, it is unclear how the "field capacity" value is determined from the knowledge of the soil hydraulic properties. Which hydraulic property are you referring to? Is it the soil-water retention function, or the hydraulic conductivity function, or both? Please clarify at which matric pressure the "field capacity" value is computed. Does this matric head value depend on the soil type? Given that your paper deals with field analysis, perhaps it would be more appropriate to determine the "field capacity" value through an actual (or, synthetic) drainage process, which accounts for the actual layering of the soil profile. Regarding the point at which "permanent wilting" occurs, further information is desirable about the matric pressure value (in MPa, not mPa) associated with specific vegetation types | Agreed, we revised model description section and REW information. Soil water content at filed capacity (and at wilting point) is defined using van Genuchten retention curve at -6.3 kPa (-1585 kPa) pressure (recommended values for Germany). Both retention and conductivity curves are soil-type dependent and parameterized for each layer from soil physical properties using Wessolek PTF. Regarding wilting point for different plants, we deleted the sentence, since these values actually referred to critical leaf potential for stomatal closure. | Section 2.3.1 and 2.3.3 |
| 2 | It would be beneficial to a wider readership to provide a more detailed comparison between point-scale and grid-scale modeling. This concern was also raised in the beginning of the Ref.#1's appraisal. I believe that your reply to this question should be more comprehensive. My feeling, but I might be wrong, is that your comparison between point-scale and grid-scale is incomplete. This is because the change-of-support problem appears to be understated or even overlooked. The failure to consider the effect of changing spatial support may result in the generation of biased comparisons. | Agreed, we elaborated on the topic in discussion section. | Section 4.1 |

**2. Review 1 comments**

| # | Comment | Answer | Changes |
|---|---------|--------|---------|
| 1 | MAJOR: The authors have developed a point-scale modelling system and have emphasised throughout the paper that this type of analysis has several advantages over a grid-based modelling system. Point-scale analysis can have some advantages, but with the current availability of computing and storage facilities, grid-based analysis has become very feasible even at 30 m resolution (e.g. Vergopolan et al., 2021, https://doi.org/10.1038/s41597-021-01050-2). Furthermore, point scale analysis does not consider lateral movement of water, which is particularly important over shallow soils and sloping terrain. Therefore, grid-based systems have also advantages with respect to point-scale simulations. The discussion of the limitations of the proposed approach should be clarified and the comparison with grid-based analysis should be made more fair. | Agreed partly. We added discussion section on point vs grid based soil moisture simulations for forest areas with regard to resolution, data input, pros and cons, plausibility and user orientation. A few extra features of the point-based system which are not mentioned in the text include the following ones. For forest managers the actual data availability at the point (particularly detailed soil information and its near real time as well as retrospectacle hydrological behaviour) is much more of usage as the generalised grid-based information, everything else is often redundant. Further, point modelling has the advantage that we can update point information (input parameters) and/or add new points (soil profiles) easily and at any time without having to dismantle the entire model concept. The point approach therefore offers an open space approach for continuous improvement. Point display has more practical advantages because additional information from the site (soil profile and soil properties, climate summary, topography information) fits into the display, not to forget the helpful background (aerial, OSM, forest site maps etc). On the other hand, grid-based simulations are easily for postprocessing and comparison to other similar system outputs. Regarding grid-based soil moisture estimates from Vergopolan et al. 2021. This is a reanalysis dataset (with application of modelling and several stages of downscaling techniques), which is not a daily operational framework, | Section 4.1 |

| # | Comment | Answer | Changes |
|---|---------|--------|---------|
| | | therefore is not comparable regarding the computational powers behind. Further, data from the satellite sensors used in the study typically have a depth penetration of up to 5 cm on the clear ground and do not work in the forested areas, not to mention, that probably the setup and chosen soil parameterisation dataset did not account for the difference in humus and mineral horizons physical properties in forests. | |
| | | Yes, the used 1D model could not adequately represent lateral flows (especially inflow) on the sloped terrain and we will address it in the discussion. However, this is not so critical for the study sites, since approximately 75% of forest floors (at least in Saxony) are characterised by predominant vertical seepage water movement and here LWF BROOK90 works better than, for example, the grid-based SWAT or TOPMODEL. Finally, lateral inflows and outflows, which are the main advantage of grid-based models, have to use a good quality DTM. In Central Europe, DEMs (and therefore streamflow networks) are massively covered with artefacts due to the strong anthropogenic influence (bridges, forest roads, mining traces, dams, culverts, ditches) and would have to be extensively cleaned up in order to actually achieve the alleged accuracy of soil moisture modelling, which is problematic for dense vegetation covered areas. Finally, accurate correction for tree height to produce high-resolution DTM is possible only with laser/drone scans due to high heterogeneity of the forest stand. | |

| # | Comment | Answer | Changes |
|---|---------|--------|---------|
| 2 | MAJOR: In the abstract it reads "soil moisture monitoring framework … which addresses the main limitations and problems of the existing monitoring systems.". This is not demonstrated at all in the paper. | Agreed, elaborated in a discussion section (see also comment below). As the primary stakeholders and users of the system are forest managers of different administrative levels, we gather direct feedback from them. And they agreed that the existing systems are too coarse in resolution, which was the main point of the development of this point-based modelling system with point-based soil information data from National Forest Inventory and high-resolution local soil map (1:10 000, where the smallest area-mapping unit is about 0.5 ha), rather than generalised soil maps of coarser resolution (1:1 000 000). Nevertheless, there are hardly any directly quotable sources to be found to this problem (only 'practical experience' and 'expert knowledge') and we could only summarise and mention this feedback.
From the point of view of soil hydrology, however, there is also a need to map the high spatio-temporal heterogeneity of soil moisture, which is largely controlled by the local variability of physical soil properties, topography and vegetation. (e.g. https://doi.org/10.1029/2022WR032719 or https://doi.org/10.1016/j.foreco.2016.12.024 or https://doi.org/10.1016/j.foreco.2020.118671). Therefore, our approach is to link the data available at the plot level with the best available model parameters and modelling techniques at the same spatial scale. | Abstract, section 4.1 |

| # | Comment | Answer | Changes |
|---|---------|--------|---------|
| 3 | MAJOR: Two soil moisture modelling systems were mentioned in the introduction, the German Drought Monitor and the German Weather Service Soil Moisture Viewer. These systems have high resolution, 4 km and 1 km respectively, not so different from the point modelling system developed in this study, note that the average distance of the rain gauge is 5 km and more for the other meteorological data. I believe that a comparison should be made between the simulation carried out in this study and these systems. Through this analysis, the potential added value of the developed monitoring system can be evaluated. | Agreed partly. The detailed comparison was added in the new discussion section, while information in the introduction was reduced. All three systems use more or less the same meteorological data (main difference are quality-control and interpolation techniques). Currently, the online version of German Drought Monitor uses the MHm model, which possesses limitations with regard to evaporation representation and 1:1 000 000 resolution soil map. The German Weather Service Soil Moisture Viewer uses LWF-BROOK90 (same as in the presented setup) model for forests and a 1:1 000 000 resolution soil map. The system was updated after the manuscript submission (before it was only for croplands). It is questionable whether the average distance of 5 km to the rain gauge dictates the resolution of the product. For instance, German Drought Monitor uses regular-grid-interpolation inputs and has no variability due to soil or meteorological datasets within a grid (4 km), while the presented framework will still possess point-wise variability due to higher resolution of soil dataset and variable distance to the meteostations for each point. Exact build-up of the second system is unfortunately unknown, but presumably the above mentioned statement holds for it as well (at least regarding variability due to soil dataset). Raster-based simulation at the resolution of the used soil dataset will require a grid of 10 m, which is implausible for the implementation due to computational power requirement and technical problems | Section 4.1 |

| # | Comment | Answer | Changes |
|---|---------|--------|---------|
| | | of the result representation (in an interactive and not just raster-based way). The raw simulation data from the two systems is not open-sourced (only post-processed and converted in indexes and presented via picture-based format). We will make requests to corresponding authors of these two systems. If the data will not be available, we will be able to make only a qualitative analysis and discussion of the setups, which from our point of view will still be enough to conclude on the potential added value of the presented framework. | |
| 4 | MAJOR: A major and important issue is the lack of validation. Only one figure, not discussed, with a comparison to a single soil moisture station. In addition, the paper also reports long-term averages of evaporation per forest type. How accurate are the evaporation estimates? How accurate are the soil moisture estimates? In order to be published, the paper should perform a robust validation of the simulations. | Agreed partly, initially the main purpose of the presented paper was not about robust validation, but about presenting the system itself. Besides, it is mentioned in the text, that the soil moisture pilot version of the setup was already validated (for different forest types and grass) based on soil moisture measurements in Saxony (http://dx.doi.org/10.1127/metz/2023/1155). Additionally, we want to point out, that due to high heterogeneity of the soils in the region, direct comparison of measured soil moisture with simulations from nearby BWI point could lead to unexpected results, even if the vegetation above is completely the same, due to possible differences in soil structure and profile depth. Evaporation was validated similarly to the presented setups (with original BROOK90 model, which has the same evaporation module) for 5 eddy-covariance towers in Saxony (https://doi.org/10.5194/hess-26-3177-2022). | Section 2.2.4, Section 2.4, Section 3.1, Appendix A1, A4 |

| # | Comment | Answer | Changes |
|---|---------|--------|---------|
| | | Nevertheless, we added an extra sections to the methods and results with validation using available data of 51 stations (forest climate stations and FLUXNET/ICOS towers) and nearby simulated BWI points. | |
| 5 | MODERATE: The text contains several errors, and some parts are unclear (see specific comments for some of them). Acronyms are not defined. Many references are in German. Four figures are in German. This can't be accepted in an international journal. The figures should be translated into English and the German references should be avoided or minimised. | Agreed, unclear parts and errors mentioned below were revised, specific acronyms were checked, and their number was reduced to minimum. The platform is now illustrated with updated Figures 13, 14, and 15 (new numbering) from the new English-version of the website. | Throughout the text, Figures 13,14,15 |
| 6 | MODERATE: The model is described in Section 3.1. Several modules are mentioned, e.g. vegperiod, betamode, b90, ... However, a detailed description of these modules is missing. The reader is lost and, for example, it's unclear how the values of the model parameters were assigned. How many parameters? The model description needs to be improved. | Agreed partly, the main principles of mentioned modules are presented in the text (L150-152 and L161-166). We will revise and enhance to a certain extent the description, however providing a full description of them in the text is unnecessary from our point of view, as the potential reader could always refer to a provided reference and it shifts the focus of the section. Additionally, we suggest putting a 'summary' table on the model parameterization in the Appendix. | Section 2.3.1, Appendix A3 |
| 7 | L71: "with an operational climate data". It's not climate, but meteorological data. To be corrected throughout the paper. | Agreed, was corrected and checked throughout the text. | Section 1 |
| 8 | Figure 1 caption: Specify acronyms. | Agreed, added. | Figure 1 |

| # | Comment | Answer | Changes |
|---|---------|--------|---------|
| 9 | L92: The size of the investigated area should be specified. | Agreed, added. It is about 55000 km2. | Section 2.1 |
| 10 | L128: What is the "REST-API access"? | Agreed, clarified. | Section 2.2.3 |
| 11 | L135-136: The sentence is unclear, and it should be revised. | Agreed, revised. | Section 2.2.2 |
| 12 | L140: What does LWF stand for? Check all acronyms. | Agreed, added. LWF stands for 'Landesanstalt für Wald und Forstwirtschaft' Bayern, where the original model was first modified. | Section 2.3.1 |
| 13 | L188: Why are some stations filtered? What do the authors mean with "filtered"? | Agreed, clarified. Station list available for the whole of Germany was reduced based on spatial (study region plus 30 km buffer zone) and temporal (data availability within the last 10 years period) principles. | Section 2.3.2 |
| 14 | L190: Which criteria should be matched? | | Section 2.3.2 |
| 15 | L191-193: The sentence is unclear, and it should be revised. | Agreed, clarified. | Section 2.3.2 |
| 16 | L201: It's not clear for which period the soil moisture data are simulated from the system. Here it reads 5 months, later in the text 30-year period. It must be clarified. | Agreed, clarified. In operational mode we simulate 10 years each day. 30-years simulations were done once to calculate quantile REW values for each plot. Here we mean, that tests of the meteorological data retrieval for 10-year simulations during the June-October 2023 period did not show large deviations in station data availability. This means that up-to-date data for each used variable is almost stable regarding the number of stations. | Section 2.3.2 |
| 17 | L08-209: Approximate distance for rain gauge equal to 5 km, more than 10 km for other meteo data. The actual resolution of the simulated soil moisture cannot be less than 5 km. | Agreed partly, see comment #3. The resolution of the meteorological forcing does not necessarily dictate the resolution of the soil moisture simulations. We elaborated | Section 2.3.2 |

| # | Comment | Answer | Changes |
|---|---------|--------|---------|
| | | on it in text. For each simulated point within the same BWI plot, meteorological input will be different due to different distances, even if the original station data picked by the filter is the same. Moreover, each point has its own geographical features, including topography and soils. These two factors bring enough variability and confirm the higher output resolution compared to grid-based products. Furthermore, based on our previous study for the evaporation component, it was found that the variability and uncertainty of the parameterization datasets is higher than the uncertainty of the meteorological data (https://doi.org/10.5194/hess-26-3177-2022). | |
| 18 | L226-227: It's unclear if NA values are present or not in the data. Please revise the sentence. | Agreed, clarified. So far we did not experience such a case, while for each plot
1) stations in 30 km buffer with more than 5% on NA are removed
2) typically more than one station for plot inside the buffer
3) for 1-day-lag data 30 km buffer is expandable if NA appear
So typically after all these steps there is no NA data inside. However, if it happened (i.e. for wind or sunshine duration data, as the station network is sparse; or there is an update failure by certain DWD stations) this simplified gap-filling algorithm was added out of precaution, so 100% no NA meteorological data appear in the forcing. | Section 2.3.2 |

| # | Comment | Answer | Changes |
|---|---------|--------|---------|
| 19 | L233-235: The sentence is unclear, and it should be revised. | Agreed, revised. | Section 2.3.2 |
| 20 | L270: I don't agree that raster-based simulations do not account for local conditions, it depends on the grid size. Please revise (see the first general comment). | Agreed, revised and elaborated is discussion section 4.1. | Section 4.1 |
| 21 | L273-274: I don't believe the two examples "illustrate the advantage of the point-based framework". This part should be revised. | Agreed, the section was renamed to 'Effect of the local scale on the soil moisture under drought conditions' | Section 3.2 |
| 22 | L335: What is the time period of the long-term simulations? | Agreed, added. Time-period of long-temp simulations is 1990-2024 and 30-year time period for the referred figure is 1991-2020. | Section 2.3.3, Section 3.4 |

**3. Review 2 comments**

| # | Comment | Answer | Changes |
|---|---------|--------|---------|
| 1 | Improve readability: there are multiple acronyms throughout the text (and also in the figures) which are not always easy to remember (reader needs to search through the document). Some of the explanations are very specific for forest managers or forest researcher (e.g. see my comment on track corners). I think the soil moisture monitoring system you have developed is valuable and very interesting. | Agreed, specific acronyms were reduced to minimum. | Throughout the text |
| 2 | Scientific discussion and placing your SM monitoring within the context of available literature and ongoing SM monitoring efforts elsewhere: I miss more discussion of how the system you present is "an operational high-resolution soil moisture monitoring framework for the forests in Middle Germany, which addresses the main limitations and problems of the existing monitoring systems". What are the existing monitoring systems and what are their shortcomings? Could you include a discussion on that? Also, are you referring to existing monitoring systems in Germany or around the world? What would be a comparably good SM monitoring system in another country? An example of an SM monitoring systems, based on SM observations from cosmic ray neutron sensing would be COSMOS UK (see https://cosmos.ceh.ac.uk/data ). They feature a similar system to the traffic light system you describe, you could perhaps compare it to such a system and/or include examples from forested sites. | Agreed. Existing systems we found are briefly mentioned in the introduction. The more detailed comparison was added in the new discussion section, while information in the introduction was reduced, as also requested by Reviewer #1. We are referring mainly to European systems, but we will continue our search to find, mention and compare similar systems worldwide. Regarding the COSMOS UK system you've mentioned. It is a very nice example of a monitoring system using observations. We mentioned it in comparison. However, as the observations are very sparse, they are unable to cover larger areas as well as to be representative with regard to spatial soil heterogeneity. And coupled with absence of additional information about the site itself, it is hardly applicable for the forest management. | Section 1, Section 4.1 |

| # | Comment | Answer | Changes |
|---|---------|--------|---------|
| 3 | Discussion on soil moisture observations and their usefulness in such a SM monitoring framework: There is no mention or discussion on how to incorporate actual soil moisture observations in your framework and how it would benefit from it. Are there any hydrological observatories where you could apply your modelling framework but then improve it? You make a shy suggestion in the Outlook section, but that is rather short and underdeveloped. | Agreed, elaborated. At the present, operational soil moisture observations are available only for Saxony (forest climate stations, measurements under grass vegetation), which are already pre-processed and delivered through a third-party data provider (Pikobytes GmbH). Unfortunately, we don't have access to similar operational soil moisture data (forest climate stations) in Thuringia and Saxony-Anhalt to integrate it in the framework. We are still trying to negotiate it. In general, every similar system will definitely profit from coupling its simulations with observations, where it is possible. This helps not only to validate it 'on-fly', but in long-term to improve parameterization. | Section 4.2 |
| 4 | Perhaps too much German on figures and in text: in Section 4.4. as a non-proficient German speaker I found it difficult and unmotivating to follow. I would find it much more interesting, if the platform could be presented in the publication already with translation in English (i.e. wait until then to publish this contribution or state a date, ideally in the near future, when the website will be available in English). Alternatively you can take the focus away from the online platform and mention it briefly and also produce a short video tutorial in English for users interested in the data and science behind it. Then focus much more on discussing the science (see general comment ii). | Agreed, we shifted focus slightly from the web-platform description more in the framework description. Further, added a validation section in results and discussion section on the comparison of our platform to other existing systems. Nevertheless, we added an English version of the website and plan to record tutorials for potential English-speaking users. Despite the fact, that the main users of the system are German-speaking forest managers, we believe the platform itself and the produced data could be of interest to the international community: both national forest authorities (i.e. to learn and build similar systems) | Section 2.3.3, Section 2.4, Section 3.1 |

| # | Comment | Answer | Changes |
|---|---------|--------|---------|
| 5 | If the platform is intended to engage more stakeholders and forest managers in Germany, I believe it would be much more beneficial to publish in a German scientific journal which is also easily available for environmental authorities and forest managers. This is also where German speaking scientists interested in the platform (again because it is only in German at the moment) can explore it. At the moment the only way to go through the different options is via translating the page. When you do that the images stall and the page takes longer to load. For the expert mode you need to know at least some German or be patient to translate to start using the data files downloaded. | and scientific community (i.e. in research of forest ecosystems dynamics and variability). The pilot version of the system was already presented for German-speaking users in parallel via various application-oriented journals and internet platforms. | |
| 6 | To address such a public, perhaps also the text would need to be rewritten and more emphasis on how to use the platform and perhaps a couple of examples of the benefits of using it (i.e. practical examples) should be included. | Agreed, we elaborated on it in order to address the practical part, with examples from forest managers. | Section 3.5, Section 4.1 |

| # | Comment | Answer | Changes |
|---|---------|--------|---------|
| 7 | Figure 1: Add a small inlet of Germany in one of the corners. Complement the Figure 1 caption with the meaning of the 3206 BWI abbreviation to aid readers. Briefly explain what the black dots mean (I understand is the inventory but please make it explicit). | Agreed, corrected and added. | Figure 1 |
| 8 | Line 128: Have a very brief explanation of what REST API access is (few words) | Agreed, added. | Section 2.2.3 |
| 9 | Line 139: Section 3.1. is well documented/ choices well explained. However, I suggest a more intuitive sub-header starting with the model type and then introducing the name. spell out that it is a soil hydrological model and it is 1D. Line 140: I would start the paragraph with saying what the model is about and then go into these details for the benefit of readers who are not familiar. | Agreed, corrected. | Section 2.3.1 |
| 10 | Figure 2: explain in the legend or caption what KL or RR stand for. Make the dot for the BWI sites slightly larger on the legend. Why is this figure relevant to show here and why not in Annexes? | Agreed, added. As we added more (sub)sections, text and figures to the article, we move this figure in Appendix. | Appendix A2 |
| 11 | Figure 3: same comment on the BWI dot | Agreed, corrected. | Figure 3 |

| # | Comment | Answer | Changes |
|---|---------|--------|---------|
| 12 | Figure 4: remove "violin plots with" from figure caption, it is redundant. Otherwise figure is quite informative | Agreed, corrected. | Figure 4 |
| 13 | Figure 5: useful figure giving a good overview. Small detail on caption, change to "for a selected" | Agreed, corrected. | Figure 5 |
| 14 | Figure 6: green balloon "Daily meteostation data from 2010" sounds like the data is from that year. I understand it is from 2010 onwards and up to current? | Agreed, it was done for space saving on the chart. We changed it to 'Meteorological data' with explanation in text, that data is from 2010 onwards. | Figure 6 |
| 15 | Section 4.1. Line 272 what does "first hundreds of meters" mean in this sentence? please rephrase or clarify. Also you could discuss the differences between the point and raster set up already in Methods (I did not see it there). This section is not so easy to read, I expect more documenting (i.e. references and comparison between raster and point set ups). | Agreed, this problem was elaborated in discussion section 4.1, as it was also pointed out by Reviewer #1. | Section 3.1 |
| 16 | Line 273 typo "in" instead of "is" | Agreed, corrected. | Section 3.1 |
| 17 | Figure 7: On your (b) plot in the legend the light yellow and green are very difficult to see. You have the same issue on the lightest colours in Figure 8 on the legend. | Agreed, we changed color schemes and increased symbol sizes in the legend to improve visibility. | Figure 9, Figure 10 |
| 18 | Fig 9: very nice and informative on the evolution of SM along a whole year | Thank you for pointing this out. | - |

| # | Comment | Answer | Changes |
|---|---------|--------|---------|
| 19 | Line 357: for the readers who may not know what track corners are, can you please include a reference? | Agreed, will be clarified in Section 2.2 | Section 2.2.3 |
| 20 | Figure 10: another interesting and useful figure from scientific point of view. | Thank you for pointing this out. | - |
| 21 | Figure 11 and 12: entirely in German, basically snapshots and (at least when I download the pdf), the resolution is quite low. I struggle to see the text (Fig 11 for example) and think it occupies unnecessary space. Instead of these figures an explanatory video could be much more useful. | Agreed the platform is now illustrated with updated Figures 13, 14, and 15 (new numbering) from the new English-version of the website. | Section 3.5 |

**4. Extra comment**

| # | Comment | Answer |
|---|---------|--------|
| 1 | We welcome the efforts to improve the existing drought monitoring systems in Germany. We wanted to draw attention to the fact that a demonstration of the 1 km version of the German Drought Monitor was presented in HESS in 2022 (https://www.doi.org/10.5194/hess-26-5137-2022). We would like to encourage the authors to add references to the 1 km version of the German Drought Monitor. | Thank you for the comment. We included this reference in the discussion section of the revised version. |